# Feature Line Embedding Based on Support Vector Machine for Hyperspectral Image Classification

Ying-Nong Chen [1,2], Tipajin Thaipisutikul [3], Chin-Chuan Han [4,*], Tzu-Jui Liu [2] and Kuo-Chin Fan [2]

1 Center for Space and Remote Sensing Research, National Central University, No. 300, Jhongda Rd., Jhongli Dist., Taoyuan City 32001, Taiwan; yingnong1218@csrsr.ncu.edu.tw
2 Department of Computer Science and Information Engineering, National Central University, No. 300, Jhongda Rd., Jhongli Dist., Taoyuan City 32001, Taiwan; ray6.ray61027@g.ncu.edu.tw (T.-J.L.); kcfan@csie.ncu.edu.tw (K.-C.F.)
3 Faculty of Information and Communication Technology, Mahidol University, 999 Phuttamonthon 4 Rd., Salaya, Nakhon Pathom 73170, Thailand; tipajin.tha@mahidol.ac.th
4 Department of Computer Science and Information Engineering, National United University, No. 1, Lienda, Miaoli 36003, Taiwan
* Correspondence: cchan@nuu.edu.tw; Tel.: +886-37-382610

**Abstract:** In this paper, a novel feature line embedding (FLE) algorithm based on support vector machine (SVM), referred to as SVMFLE, is proposed for dimension reduction (DR) and for improving the performance of the generative adversarial network (GAN) in hyperspectral image (HSI) classification. The GAN has successfully shown high discriminative capability in many applications. However, owing to the traditional linear-based principal component analysis (PCA) the pre-processing step in the GAN cannot effectively obtain nonlinear information; to overcome this problem, feature line embedding based on support vector machine (SVMFLE) was proposed. The proposed SVMFLE DR scheme is implemented through two stages. In the first scatter matrix calculation stage, FLE within-class scatter matrix, FLE between-scatter matrix, and support vector-based FLE between-class scatter matrix are obtained. Then in the second weight determination stage, the training sample dispersion indices versus the weight of SVM-based FLE between-class matrix are calculated to determine the best weight between-scatter matrices and obtain the final transformation matrix. Since the reduced feature space obtained by the SVMFLE scheme is much more representative and discriminative than that obtained using conventional schemes, the performance of the GAN in HSI classification is higher. The effectiveness of the proposed SVMFLE scheme with GAN or nearest neighbor (NN) classifiers was evaluated by comparing them with state-of-the-art methods and using three benchmark datasets. According to the experimental results, the performance of the proposed SVMFLE scheme with GAN or NN classifiers was higher than that of the state-of-the-art schemes in three performance indices. Accuracies of 96.3%, 89.2%, and 87.0% were obtained for the Salinas, Pavia University, and Indian Pines Site datasets, respectively. Similarly, this scheme with the NN classifier also achieves 89.8%, 86.0%, and 76.2% accuracy rates for these three datasets.

**Keywords:** HSI classification; feature line embedding; dimension reduction; support vector machine; generative adversarial networks

## 1. Introduction

Recently, hyper spectral image (HSI) classification has attracted the attention of researchers owing to its numerous applications such as land change monitoring, urban development, resource management, disaster prevention, and scene interpretation [1]. Generally, in HSI classification, a specific category is assigned to each pixel in an image. However, since most multispectral, ultraspectral, and hyperspectral images generate a large number of high-dimensional pixels with many category labels, it is a challenging task to effectively separate these pixels with similar land cover spectral properties. First, because

of the large number of HSI pixels, numerous schemes applied in HSI classification were based on supervised learning [1], including the support vector machine (SVM) [2,3], nearest feature line [4,5], random forest [6], manifold learning [7,8], sparse representation [9], and deep learning (DL) [1]. Second, because HSI contains high-dimensional information, many studies have focused on dimension reduction (DR) and reported the importance of DR in HSI classification [10].

Principal component analysis (PCA) [11] is the most popular DR algorithm; it subtracts the mean of the population from each sample to obtain the covariance matrix and extracts a transformation matrix by maximizing the scatter of samples. PCA also plays a role in the pre-processing for other advanced DR algorithms to remove noise and mitigate overfitting [1,10]. Linear discriminant analysis (LDA) [12], and discriminant common vectors [13] are advanced versions of PCA. Since the PCA algorithm is based on linear measurement, it is ineffective in revealing the local structure of samples when samples are distributed in a manifold structure. Many methods based on manifold learning and kernel methods have been proposed to overcome the abovementioned problem. Manifold learning was proposed to preserve the topology of the locality of training samples. He et al. [14] proposed a locality preserving projection (LPP) scheme to preserve the local topology of training data for face recognition. Because the sample scatter obtained through LPP is based on the relationship between neighbors, the local manifold information of samples is preserved, and therefore, the performance of the LPP scheme was shown to be higher than that of linear measurement methods. Tu et al. [7] presented the Laplacian eigenmaps (LE) method, which uses polarimetric synthetic aperture radar data for land cover classification. The LE scheme maintains the high-dimensional polarimetric manifold information in an intrinsic low-dimensional space. Wang and He [15] also used LPP to pre-process data for HSI classification. Kim et al. [16] presented a manifold-based method called locally linear embedding (LLE) to reduce the dimensionality of HSI. Li et al. [8,17] presented the local Fisher discriminant analysis (LFDA) scheme, which takes into account the merits of both LDA and LPP to reduce the dimensionality of HSI. Luo et al. [18] proposed a supervised neighborhood preserving embedding method to extract the salient features for HSI classification. Zhang et al. [19] employed a sparse low-rank approximation scheme to regularize a manifold structure, and HSI data were treated as cube data for classification.

Generally, these methods based on manifold learning all preserve the topology of the locality of training samples and outperform the traditional linear measurement methods. However, according to Boots and Gordon [20], nonlinear information cannot be extracted through manifold learning, and the effectiveness of manifold learning is limited by noise. Therefore, the kernel tricks were employed to obtain a nonlinear feature space and improve the extraction of nonlinear information. Because the use of the kernel tricks improves the performance of a given method [21], the kernelization approach was adopted in both linear measurement methods and manifold learning methods to improve HSI classification. Boots and Gordon [20] employed the kernelization method to alleviate the noise effect in manifold learning. Scholkopf et al. [22] proposed a kernelization PCA scheme, which makes use of kernel tricks to find a high-dimensional Hilbert space and extract non-linear salient features missed by PCA. In addition, Lin et al. [23] presented a framework for DR based on multiple kernel learning. The multiple kernels were first unified to generate a high dimensional space by a weighted summation, and then these multiple features were projected to a low dimensional space. However, using their proposed framework, they also attempted to determine proper weights for a combination of kernels and DR simultaneously, which increased the complexity of the method. Hence, Nazarpour and Adibi [24] presented a novel weight combination algorithm that was used to only extract good kernels from some basic kernels. Although the proposed idea was a simple and effective idea for multiple kernel learning, it used kernel discriminant analysis based on linear measurement for classification, and thus would not preserve the manifold topology of multiple kernels in high dimension space. Li et al. [25] proposed a kernel integration algorithm that linearly assembles multiple kernels to extract both spatial and spectral

information. Chen et al. [26] used a kernelization method based on sparse representation for HSI classification. In their approach, a query sample was represented by all training data in a generated kernel space, and all training samples were also represented in a linear combination of their neighboring pixels. Resembling the idea of multiple kernel learning, Zhang et al. [27] presented an algorithm for multiple-feature integration based on multiple kernel learning and employed it to classify HSI data; their proposed algorithm assembles shape, texture, and spectral information to improve the performance of HSI classification. In addition to obtaining a salient feature space for HSI classification, DR can be a critical pre-processing step for DL. Zhu et al. [10] proposed an HSI classification method based on 3D generative adversarial networks (GANs) and PCA. Their experimental results demonstrated that the performance of GANs is adversely affected if there is no PCA pre-processing step. However, as mentioned earlier, because PCA is a linear measurement method, it may miss some useful manifold information. Therefore, a DR algorithm that can extract manifold information should be used to improve the performance of GANs.

Finally, because of the numerous category labels of HSI, a more powerful classifier is required to improve the performance of HSI classification. Recently, DL has been viewed as the most powerful tool in pattern recognition. Zhu et al. [10] used GANs for HSI classification and obtained a favorable result. Liu et al. [28] proposed a Siamese convolutional neural network to extract salient features for improving the performance of HSI classification. He et al. [29] proposed multiscale covariance maps to improve CNN and integrate both spatial and spectral information in a natural manner. Hu et al. [30] proposed a convolutional long short-term memory method to effectively extract both spectral and spatial features and improve the performance of HSI classification. Deng et al. [31] proposed a deep metric learning-based feature embedding model that can overcome the problem of having only a limited number of training samples. Deng et al. [32] also proposed a unified deep network integrated with active transfer learning, which also overcomes this problem. Chen et al. [1] proposed fine-grained classification based on DL; a densely connected CNN was employed for supervised HSI classification and GAN for semi-supervised HSI classification.

From the presented introduction, the challenges in HSI classification can be summarized as follows:

1.  Owing to the high dimensions of HSI, an effective DR method is required to improve classification performance and overcome the overfitting problem.
2.  Owing to the numerous category labels of HSI, a powerful classifier is required to improve the classification performance.

Because DR improves HSI classification considerably and overcomes the overfitting problem, in this study, a modification of the feature line embedding (FLE) algorithm [4,5] based on SVM [33], referred to as SVMFLE, was proposed for DR. In this algorithm, SVM is first employed for selecting the boundary samples and calculating the between-scatter matrix to enhance the discriminant ability. As we know, the scatter calculated from the boundary samples among classes could reduce the noise impact and improve the classification results. Second, the dispersion degree among samples is devised to automatically determine a better weight of the between-class scatter. By doing so, the reduced space with more discriminant power is obtained. Three benchmark data sets were used to evaluate the algorithm proposed. The experimental results demonstrated that the proposed SVMFLE method effectively improves the performance of both nearest neighbor (NN) and GAN classifiers.

The rest of this paper is organized as follows: In Section 2, previous and related works are discussed. In Section 3, the proposed algorithm of SVM-based sample selection incorporated into the FLE is introduced. In Section 4, the proposed method was compared with other state-of-the-art schemes for HSI classification to demonstrate its effectiveness. Finally, in Section 5, the conclusions are drawn.

## 2. Related Works

### 2.1. Feature Line Embedding (FLE)

In this study, a novel SVMFLE DR algorithm based on FLE [4,5] and SVM [33] was proposed to reduce the number of feature dimensions and improve the performance of NN or GAN classifiers for HSI classification. A brief review of FLE, SVM, and GAN is provided in the following sections, after which the proposed algorithm is discussed. Consider $N$ $d$-dimensional training samples $X = [x_1, x_2, \ldots, x_N] \in R^{d \times N}$ with the labels $C = \{c_i\}_{i=1}^N$. These samples consist of $N_C$ land-cover classes for training. The projected samples in a low-dimensional feature space are obtained by the linear transformation $y_i = W^T x_i$, in which $W$ is a linear transformation matrix for DR. For clarity, we present the notation and definitions used throughout this paper in Table 1.

**Table 1.** List of notations.

| Notation | Definition |
|----------|------------|
| $N$ | The total number of samples |
| $d$ | The dimension of original space |
| $N_C$ | The number of classes |
| $X$, $Y$ | The set of training data in original space and transformed space |
| $x_i \in X$, $y_i \in Y$ | Training sample of the $i^{th}$ object($1 \le i \le N$) in original space and transformed space |
| $C$ | Pre-defined set of class labels |
| $c_i \in C$ | Class label of the $i^{th}$ object($1 \le i \le N$) |
| $W$ | Obtained linear transformation matrix for DR |
| $F_{m,n}$ | Feature line passes through two samples $y_m$ and $y_n$ |
| $F_{m,n}(y_i)$ | A projected point on feature line $F_{m,n}$ for sample $y_i$ |
| $f_{m,n}(y_i)$ | A weight (being 0 or 1) represents the connection between point and the feature line $F_{m,n}$ |
| $A$ | Affinity matrix |
| $H$ | A matrix that represents the column sums of the affinity matrix $A$ |
| $L$ | Laplacian matrix that $L = H - A$ |
| $k_1$ | The number of NFLs within the same class of point $x_i$ |
| $k_2$ | The number of NFLs owned by varied classes of point $x_i$ |
| $V$ | A matrix that maximizes the margin between different classes |
| $v_i$ | The $i^{th}$ column of $V$ |
| $G$ | Generative network |
| $D$ | Discriminator model |
| $p$ | Data distribution |
| $z$ | A random vector takes a normal distribution |
| $E$ | Expectation operator |

FLE is a DR algorithm based on manifold learning. The training sample scatters are represented by a Laplacian matrix to preserve their local structure by applying the strategy of point-to-line. In general, the main objective in FLE is to measure the distance between the sample $y_i$ and its' projected point $F_{m,n}(y_i)$ on the feature line $F_{m,n}$ that passes through points $y_m$ and $y_n$. The FLE objective function is defined and minimized as follows:

$$
\begin{aligned}
O &= \sum_i \left( \sum_{i \ne m \ne n} y_i - F_{m,n}(y_i)^2 f_{m,n}(y_i) \right) \\
&= \sum_i \left\| y_i - \sum_j M_{i,j} y_j \right\|^2 \\
&= \mathrm{tr}\left( Y(I - M)^T (I - M) Y \right) = \mathrm{tr}(W^T X (H - A) X^T W) \\
&= \mathrm{tr}(W^T X L X^T W).
\end{aligned}
\tag{1}
$$

Here, $y_i = W^T x_i$. The projected point $F_{m,n}(y_i)$ is represented by a linear combination of $y_m$ and $y_n$ as $F_{m,n}(y_i) = y_m + t_{m,n}(y_n - y_m)$, $t_{m,n} = (y_i - y_m)^T (y_m - y_n) / (y_m - y_n)^T (y_m - y_n)$,

and $i \neq m \neq n$; therefore, the discriminant vector from sample $y_i$ to its' projected point $F_{m,n}(y_i)$ is given as $y_i - \sum_j M_{i,j}y_j$ based on some basic algebraic operations. The weight $f_{m,n}(y_i) = 1$, if feature line $F_{m,n}$ is chosen; being 0, otherwise. When the weight $f_{m,n}(y_i) = 1$, two elements in the $i^{th}$ row in matrix $M$ are obtained as $M_{i,m} = t_{n,m}$, $M_{i,n} = t_{m,n}$, $t_{n,m} + t_{m,n} = 1$, and the other elements in the $i^{th}$ row are set to 0 if $j \neq m \neq n$. Then, in Equation (1), our point-to-line strategy is implemented by summing all training samples to their nearest projected points on the feature line, referred to as nearest feature lines (NFLs), and is represented as $tr(W^T XLX^T W)$, where $L = H - A$ and matrix $H$ represents the column sums of the affinity matrix $A$. Based on the summary of Yan et al. [34], matrix $A$ is represented as $A_{i,j} = (M + M^T - M^T M)_{i,j}$ when $i \neq j$, and is 0 otherwise. Matrix $L$ in Equation (1) is represented as a Laplacian matrix. Refer to [34] for more details.

Considering the supervised FLE, the label information is used, and two parameters $k_1$ and $k_2$ for obtaining the within-scatter matrix $SSW_{FLE}$ and between-scatter matrix $SSW_{FLE}$ are manually determined:

$$SSW_{FLE} = \sum_{k=1}^{N_C} \left( \sum_{x_i \in c_k} \sum_{F_{m,n} \in F_{k_1}(x_i,c_k)} (x_i - F_{m,n}(x_i))(x_i - F_{m,n}(x_i))^T \right) \tag{2}$$

$$SSB_{FLE} = \sum_{k=1}^{N_C} \left( \sum_{x_i \in c_k} \sum_{l=1,l\neq k}^{N_C} \sum_{F_{m,n} \in F_{k_2}(x_i,c_l)} (x_i - F_{m,n}(x_i))(x_i - F_{m,n}(x_i))^T \right) \tag{3}$$

$F_{k_1}(x_i, c_k)$ describes the set of $k_1$ NFLs within the same class, $c_k$, of a specified point $x_i$, i.e., $f_{m,n}(y_i) = 1$, and $F_{k_2}(x_i, c_l)$ is a set of $k_2$ NFLs owned by varied classes of point $x_i$. Then, the Fisher criterion $tr(W^T SSB_{FLE}W/W^T SSW_{FLE}W)$ is used to maximize and obtain the transformation matrix $W$, which consists of eigenvectors with the corresponding highest eigenvalues. In the final step, a new projected sample in the low-dimensional feature space is described by the linear projection $y = W^T x$, and an NN or a GAN classifier is used for classification later.

## 2.2. Support Vector Machine (SVM)

SVM is used for binary classification; $c_i \in \{-1, 1\}$. It creates the hyperplane with the maximum margin to separate two classes. The optimization problem is solved as follows to obtain the maximum margin model with parameters $v \in R^d$ and $b \in R$:

$$\min_{v,b,\xi} \frac{1}{2}\|v\|_2^2 + T\sum_{i=1}^{N} \xi_i, \text{ s.t. } c_i\left(v^T x_i + b\right) \geq 1 - \xi_i, \ \xi_i \geq 0, \ i = 1, \ldots, N \tag{4}$$

where $\xi = \{\xi_i\}_{i=1}^N$ are the slack variables and $T \geq 0$ is used as the regularizer for handling the trade-off between the classification error and maximum margin. Moreover, SVM could also be extended to a multi-class version. Assume a training set $(X, C)$, where $c_i \in C$ is the data label and $c_i \in \{1, \ldots, N_C\}$. The decision boundary for the $j^{th}$ class is the weighted sum of support vectors $v_j \in R^d$ with a scalar bias $b_j \in R$, where $j = 1, \ldots, N_C$ indicates the various classes. Then, a multi-class SVM is used to solve the following optimization problem:

$$\min_{V,\xi} \frac{1}{2}\sum_{j}^{N_C} v_j^T v_j + T\sum_{i=1}^{N} \xi_i \tag{5}$$

$$\text{s. t. } v_{c_i}^T x_i - v_j^T x_i \geq e_i^j - \xi_i, \ \xi_i \geq 0, \ i = 1, \ldots, N, \ j = 1, \ldots, N_C \tag{6}$$

## 2.3. Generative Adversarial Networks (GAN)

GAN originated from the concept of game theory and provided an alternative representation of the maximum likelihood method. Two major components, a generator $G$

and a discriminator $D$, compete with each other during the training process to fulfill the Nash equilibrium. According to [10], in order to learn the generator $G$'s data distribution $p_G$ from data $x$, the real samples are under a distribution $p(x)$ and the variable of input noise is under a priori $p(z)$. That means the random noise $z$ is the input for the generator, then the generator produces a mapping space $G(z)$. The discriminator $D(x)$ identify $x$ as a real sample or not from training data by probability. In the training process, generator $G$ is trained to minimize $\log(1 - D(G(z)))$ while discriminator $D$ is trained to maximize $\log(D(x))$. Therefore, the objective function is to solve the following:

$$\min_{G} \max_{D} V(D, G) = E_{x \sim p(x)}[\log(D(x))] + E_{z \sim p(z)}[\log(1 - D(G(z)))], \tag{7}$$

in which $E$ is the expectation operator. In order to keep the generator's proper gradients when the discriminator's classification accuracy is high; the loss function of the generator is modified to maximize the probability of identifying a sample as real instead of minimizing the probability of it to be identified as false [10].

To make sure that the generator has proper gradients when the classification accuracy for the discriminator is high, the generator's loss function is usually formulated to maximize the probability of classifying a sample as true instead of minimizing the probability of it to be classified as false [28]. Hence, the formulated loss function can be rewritten as follows:

$$\min_{G} V(D, G) = -E_{z \sim p(z)}[\log(D(G(z)))] \tag{8}$$

In addition, the GAN classifier in [10] takes both spectral and spatial features into account for HSI classification. It can achieve a better performance than spectral features only. For more details about the GAN, refer to [10].

## 3. Feature Line Embedding Based on Support Vector Machine (SVMFLE)

Hyperspectral images are the sensing data with high dimensions. Due to the high dimensional properties, the original pixel dimensions are first reduced by the PCA process. Generally, PCA can find the transformation matrix for the best representation in the reduced feature space. Moreover, PCA also solves the small sample size problem for the further supervised eigenspace-based DR methods, for example, LDA [12], LPP [14], . . . , etc. The within-class and between-class scatters are two essential matrices in eigenspace-based DR. The FLE-based class scatters have been shown to be the effective representation in HSI classification [4]. Selecting the discriminant samples is the key issue in scatter computation. In [4], the selection strategy, the first $k$ nearest neighbors for a specified sample, is adopted to preserve the local topological structure among samples during training. In this proposed DR method as shown in Figure 1, the samples with more discriminant power, that is, support vectors (SV), are selected for enhancement. SVMFLE calculates the between-class scatter using the SVs which are found by SVM. According to our experience, the scatter calculated from the boundary samples among classes could reduce the noise impact and improve the classification results.

In the proposed framework as shown in Figure 1, all pixels on HSI are employed to obtain a transformation matrix $W_{PCA}$, and projected into a low dimensional PCA feature space. Next, some training samples are randomly selected, and SVs between classes are extracted by SVM in the PCA feature space. After that, FLE within-class scatter matrix $SSW_{FLE}$, FLE between-scatter matrix $SSB_{FLE}$, and SV-based FLE between-class scatter matrix $SSB_{SV}$ are calculated to maximize Fisher's criterion for obtaining the transformation matrix $W^*$. The final projection matrix is $W = W_{PCA}W^*$. Then, all pixels on HSI are projected into a five-dimensional feature space by the linear transformation $y_i = W^T x_i$. Next, 200 training samples of dimension five in the SVMFLE feature space were randomly selected to train a GAN classifier. Finally, all pixels on HSI in this SVMFLE feature space are used for evaluating the performance of the GAN classifier.

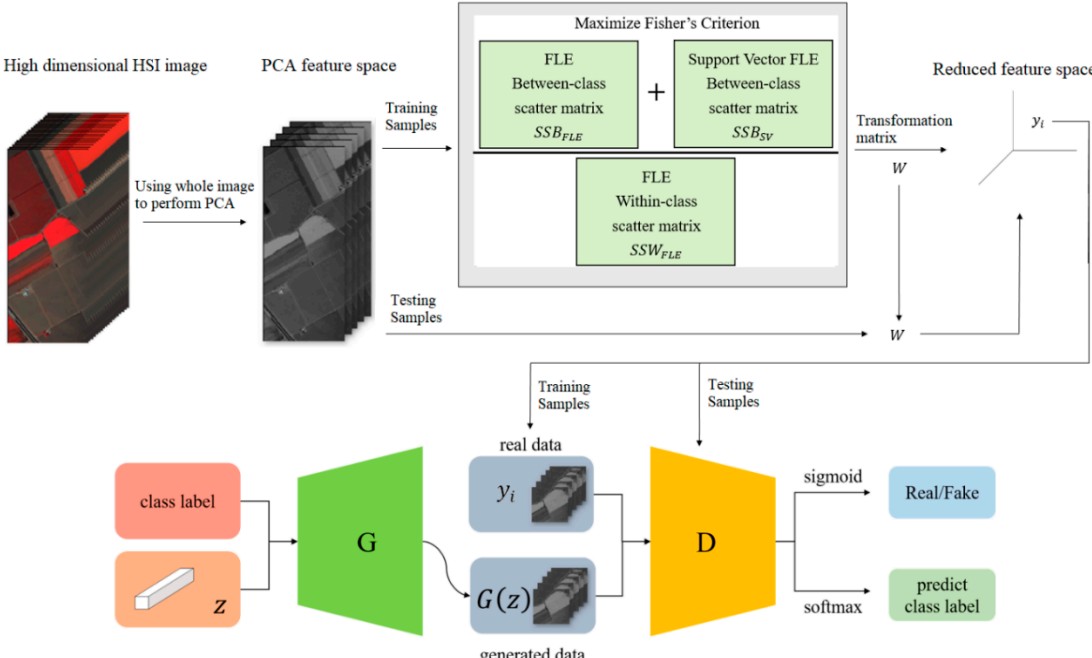

**Figure 1.** The framework of SVMFLE DR with a GAN classifier.

The within-class scatter $SSW_{FLE}$ is calculated from the discriminant vectors of the same class by using the FLE strategy. Consider a specified sample $x_i$, the discriminant vectors are chosen from the first $K_1$ nearest features lines generated by its eight nearest neighbors of the same class as shown in Equation (2). On the other hand, the between-class scatter is obtained from two parts. One scatter $SSB_{FLE}$ is similar to the approach in [4]. The first $K_2$ nearest feature lines are selected for the between-class scatter computation from six nearest neighbors of sample $x_i$ with different class labels as shown in Equation (3). Here, parameters $K_1$ and $K_2$ are set as 24 and 12, respectively, in the experiments. The other part of the between-class scatter is calculated from the support vectors generated by SVM. The one-against-all strategy is adopted in SVM, for example, two-class classification. Consider the positive training samples of a specified class $c$ and the negative samples of the other classes. These training samples are inputted to SVM for a two-class classification. After the learning process, as we know, the decision boundary is determined from the weighted summation of support vectors. These support vectors are the specified samples near the boundaries among classes. Two SV sets, positive SV $u_c^{pos}$ and negative SV $u_c^{neg}$ sets, are obtained to calculate the between-class scatter $SSB_{SV}$ as shown in the following equation:

$$SSB_{SV} = \sum_{c=1}^{N_c} \left( \sum_{x_i \in u_c^{pos}} \sum_{F_{m,n} \in u_c^{neg}} (x_i - F_{m,n}(x_i))(x_i - F_{m,n}(x_i))^T \right) \tag{9}$$

These two scatters $SSB_{FLE}$ and $SSB_{SV}$ are integrated as:

$$SSB_{SVMFLE} = \alpha SSB_{SV} + (1 - \alpha)SSB_{FLE}, \; 1 > \alpha > 0. \tag{10}$$

Here, parameter $\alpha$ indicates the ratio between the scatter for all points $SSB_{FLE}$ and the scatter for support vectors $SSB_{SV}$. Since support vectors usually locate at the class boundary regions, they are the samples with more discriminant power for learning. The transformation matrix $W$ is found by maximizing the Fisher criterion $\mathrm{tr}\left(W^T SSB_{SVMFLE} W / W^T SSW_{FLE} W\right)$, in which matrix $W$ is composed of the eigenvectors with the corresponding largest eigenvalues. The projected sample in the low-dimensional space is calculated by the linear projection $y = W^T x$. Furthermore, the reduced data were used to train the GAN classifier in [10].

The reduced HSI pixels are tested by the discriminator $D$ in GAN. The pseudo-codes of the SVMFLE DR algorithm are listed in Table 2.

**Table 2.** The algorithm of SVMFLE DR.

| Input: | A $d$ -dimensional training set $X=[x_1,x_2,\ldots,x_N]$ consists of $Nc$ classes. |
|---|---|
| **Output:** | The transformation matrix $W$. |
| Step 1: | PCA projection: Perform PCA to obtain the transformation matrix $W_{PCA}$ using the whole HSI pixels. All HSI pixels are projected from a high-dimensional space to a low dimensional space using matrix $W_{PCA}$, a PCA feature space. |
| Step 2: | Randomly select the training samples from HSI in the PCA feature space. |
| Step 3: | The support vectors between classes are obtained by applying the SVM. |
| Step 4: | Calculate a within-class matrix and two between-class matrices using Equations (2), (3), and (9), respectively. |
| Step 5: | Integrate two between-class matrices using Equation (10). |
| Step 6: | Fisher's criterion maximization: The Fisher's criterion is maximized to obtain the transformation matrix $W^* = arg\ maxSSB_{SVMFLE}/SSW_{FLE}$, which is composed of $\gamma$ eigenvectors with the largest eigenvalues. |
| Step 7: | Output the projection matrix: $W = W_{PCA}W^*$. |

Next, an indicator is defined to determine the better value $\alpha$ in Equation (10) by measuring the overlapping degree among class samples. Consider a sample $x$ in a specified class $C_i$, the Euclidean distances from every sample $x$ to its corresponding class mean $\mu_i$ are summed to represent the dispersion degree of samples as defined below:

$$SSE_i = \sum_{x \in C_i} dist(x, \mu_i) \tag{11}$$

It is defined to be the within-class distance for class $C_i$. On the other hand, the total Euclidean distance from every sample to the population mean μ is also calculated as follows:

$$SSE_{Total} = \sum_{x \in X} dist(x, \mu) \tag{12}$$

The dispersion index $r$ is thus calculated and defined as the ratio between the summation of within-class distances and the population distance as follows.

$$r = \frac{\sum_{i=1}^{N_C} SSE_i}{SSE_{Total}}. \tag{13}$$

A toy example is given to present the dispersion degree of samples. Three class samples were randomly generated in the normal distribution forms $\mathbb{N}_1((0,0),\ 50)$, $\mathbb{N}_2((0,0),\ 30)$, and $\mathbb{N}_3((0,0),\ 40)$ as shown in Figure 2a. All three class centers are located at point (0,0) and samples are distributed in various standard derivation values. The dispersion index $r = 1$ was calculated. Similarly, another toy example is given in Figure 2b. Three classes were generated in the distributions $\mathbb{N}_1((2,3),\ 50)$, $\mathbb{N}_2((200,300),\ 30)$, and $\mathbb{N}_3((-150,300),\ 40)$. The class means located at three dispersive points that are far away from each other. The standard derivations are the same values as those in Figure 2a. The dispersion index $r = 0.24$ is also calculated. From Figure 2, the smaller dispersion index $r$, the larger dispersion between classes.

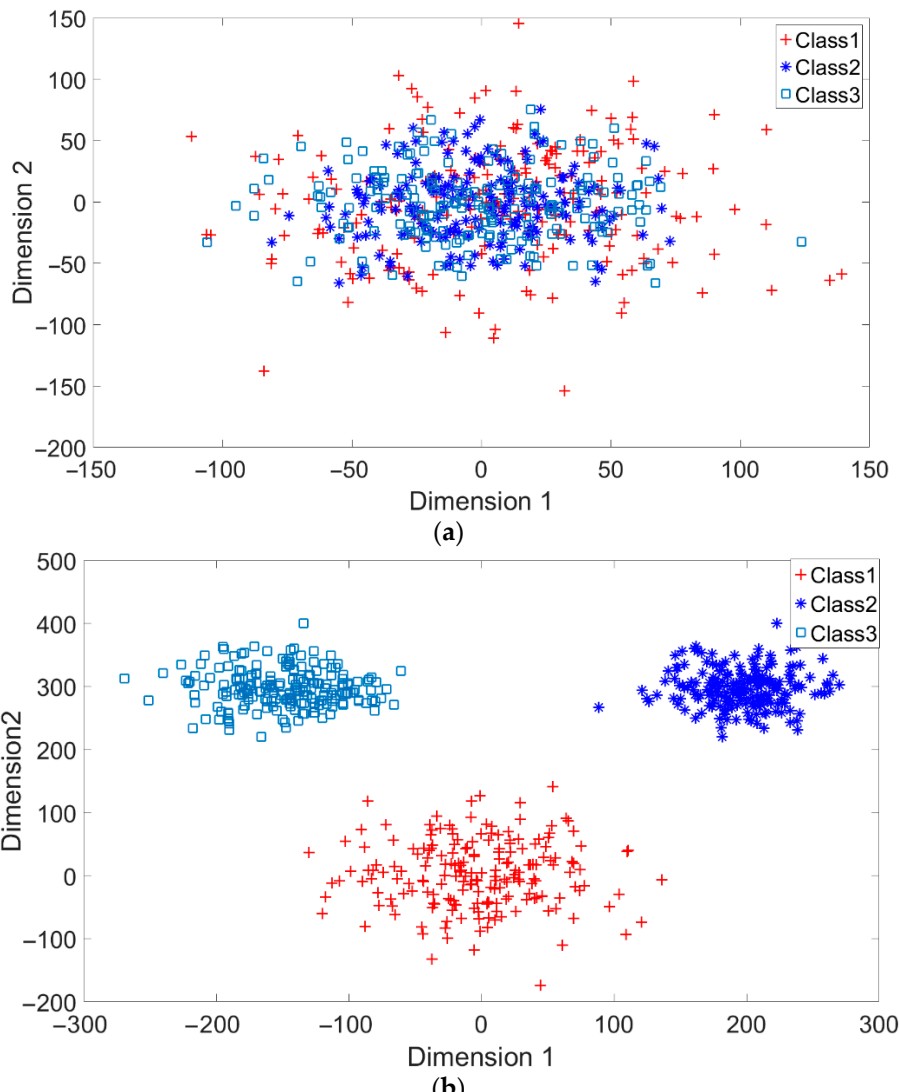

**Figure 2.** Two toy examples of overlapping: (**a**) High overlapping degree, *r* = 1, and (**b**) Low overlapping degree, *r* = 0.24.

## 4. Experimental Results

### *4.1. Measurement Metrics*

In this section, some experiments were conducted for the performance evaluation of the proposed SVMFLE DR algorithm in HSI classification. Three HSI benchmark datasets, Salinas, Pavia University, and Indian Pines Site, were given for evaluation.

Before the presentation of experimental results, three metric indices, overall accuracy (OA), class averaged accuracy (AA), and the Kappa value, are defined for performance evaluation in the following. The OA index is first calculated from the ratio of the correct predictions over the total predictions. Second, the AA index is defined as the averaged value of all class accuracies. The AA index presents the ability to classify different classes of data. The last index, the Kappa value is a consistency indicator for the classification model. The Kappa index is calculated from a confusion matrix $P^{N \times N}$. These three indices are, respectively, defined in Equations (14)–(16).

$$\text{Overall accuracy}(\text{OA}) = \frac{Corrected\ prediction}{Total\ prediction} \tag{14}$$

$$\text{Average accuracy}(AA) = \frac{1}{N_c} \sum_{i=1}^{N_c} \frac{Corrected\ prediction\ in\ class\ i}{Total\ prediction\ in\ class\ i} \tag{15}$$

$$\text{Kappa} = \frac{Q_0 - Q_1}{1 - Q_1}$$
$$\text{where } Q_0 = \frac{\sum_i P_{ii}}{\sum_i \sum_j P_{ij}} \text{ and } Q_1 = \frac{\sum_i \left(\sum_j P_{ij} \times \sum_j P_{ji}\right)}{\left(\sum_i \sum_j P_{ij}\right)^2} \tag{16}$$

### 4.2. Classification Results of Dataset Salinas

In this sub-section, the classification results on dataset Salinas are shown by comparing the proposed method with the state-of-the-art methods. The Salinas image was captured from the 224-band AVIRIS (Airborne Visible/Infrared Imaging Spectrometer) sensor over Salinas Valley in California. This HSI is composed of $512 \times 217$ pixels with 204 bands after removing 20 water absorption bands. A false-color IR image and its corresponding class label map are shown in Figure 3. In the experiment, 16 land-cover classes and a total of 54,129 pixels for dataset Salinas are tabulated in Table 3.

In the training phase, all samples in the Salinas dataset were trained to construct the PCA feature space, and 300 samples of each class, for example, 4800 samples, were randomly chosen to generate the transformation matrix for SVMFLE-based DR. To determine the better value $\alpha$ in Equation (10), the dispersion indices versus various $\alpha$ values, for example, in a range [0, 1], were calculated for the training samples as shown in Figure 4a. The index values are 0.35 if value $\alpha$ is smaller than 0.85. On the other hand, if value $\alpha$ is larger than 0.85, the indices increase. The dispersion index is smallest when $\alpha = 1.0$. These training samples were also projected into a 2D plane to show the dispersion degree of samples. Their dispersion indices are 0.41, 0.34, 0.18, and 0.07 in PCA, $\alpha = 0.0$, $\alpha = 0.95$, and $\alpha = 1.0$ conditions, respectively. From those classes drawn in the circles in Figure 4, the classes in case $\alpha = 1.0$ are separated far away from each other than those in the other cases.

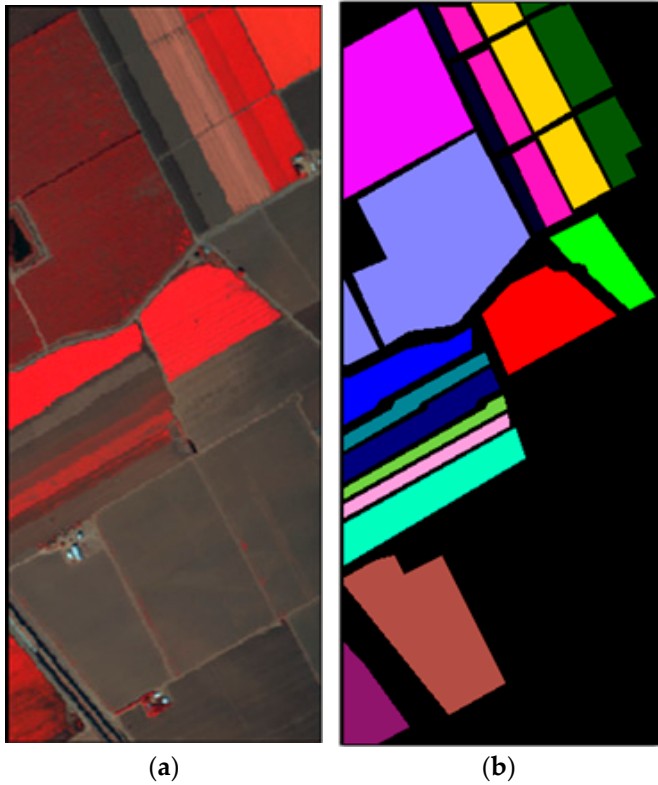

(**a**)      (**b**)

**Figure 3.** Dataset Salinas: (**a**) False color image; (**b**) Class label map.

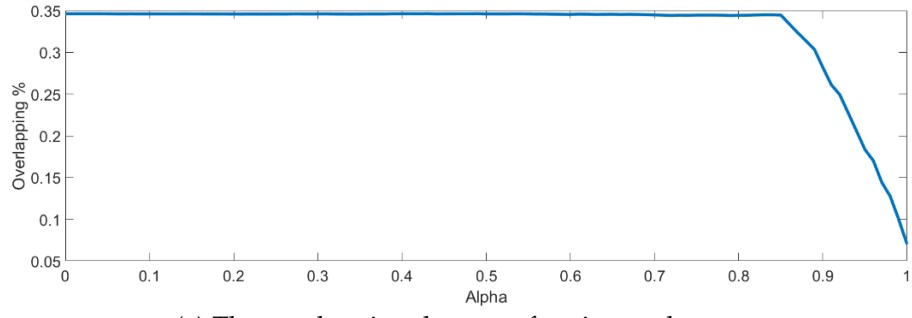

(**a**) The overlapping degrees of various values α.

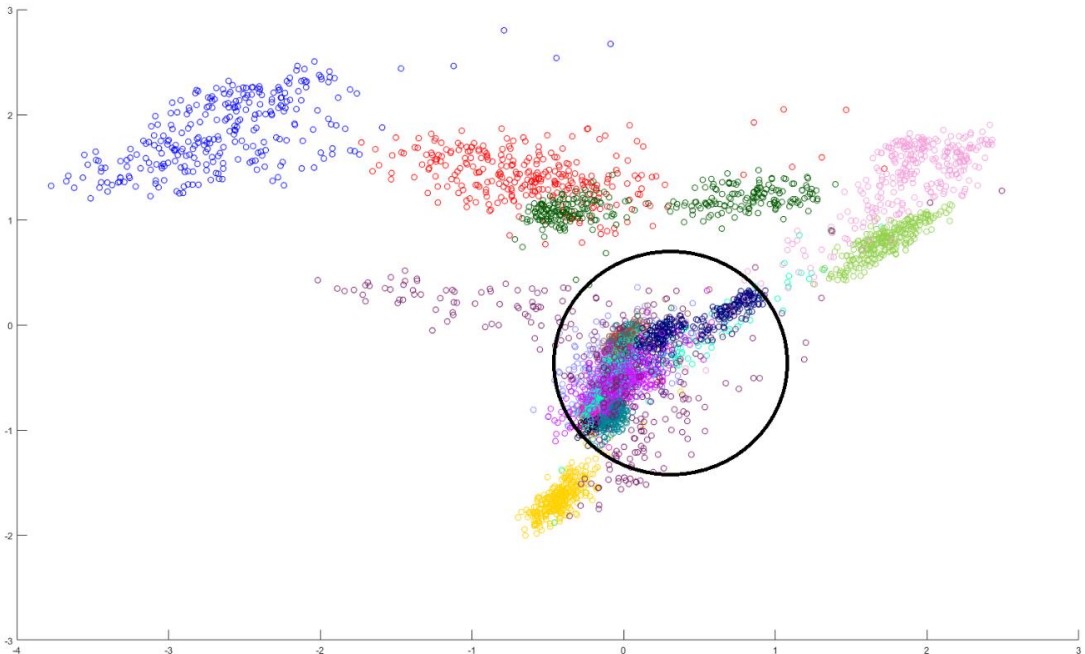

(**b**) The visualized training sample distribution of PCA and $r = 0.41$.

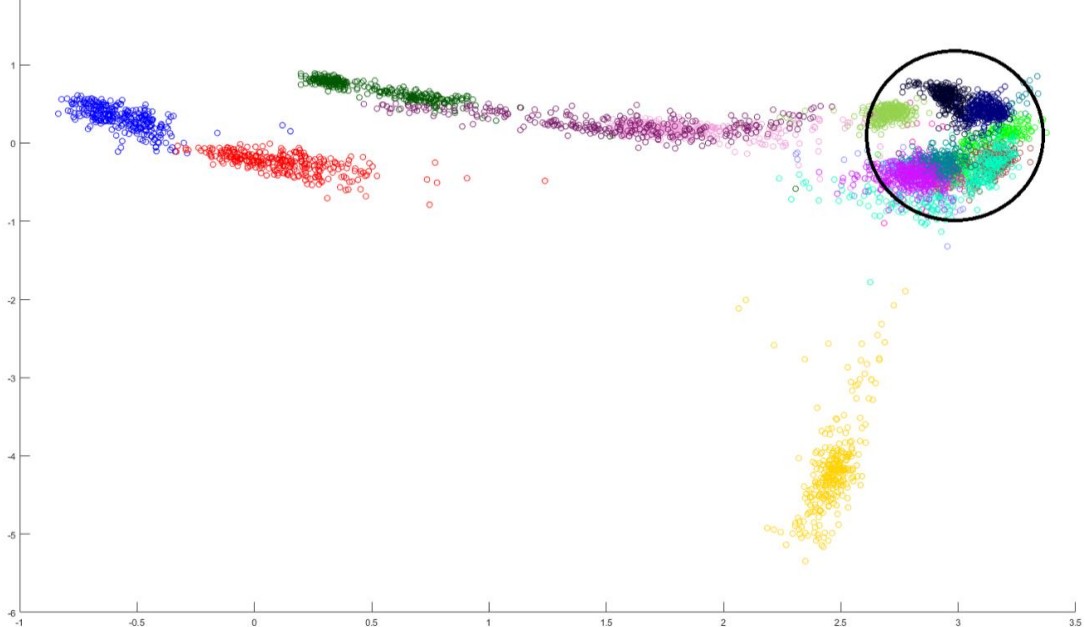

(**c**) The visualized training sample distribution of value $\alpha = 0$ and $r = 0.34$.

**Figure 4.** *Cont.*

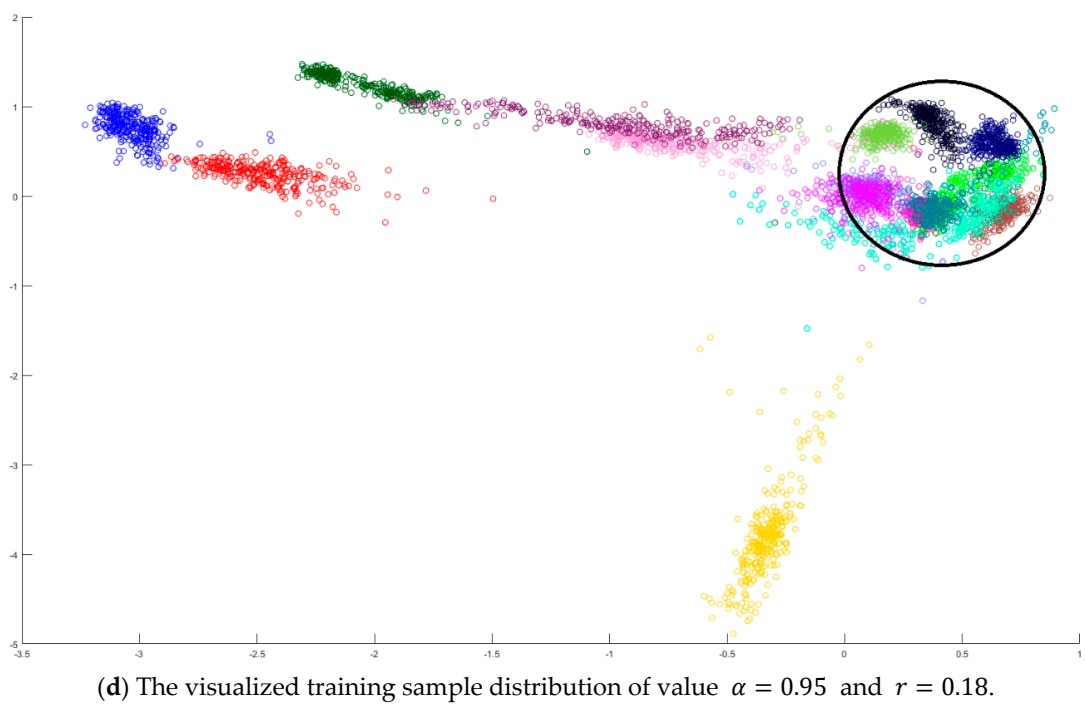

(**d**) The visualized training sample distribution of value $\alpha = 0.95$ and $r = 0.18$.

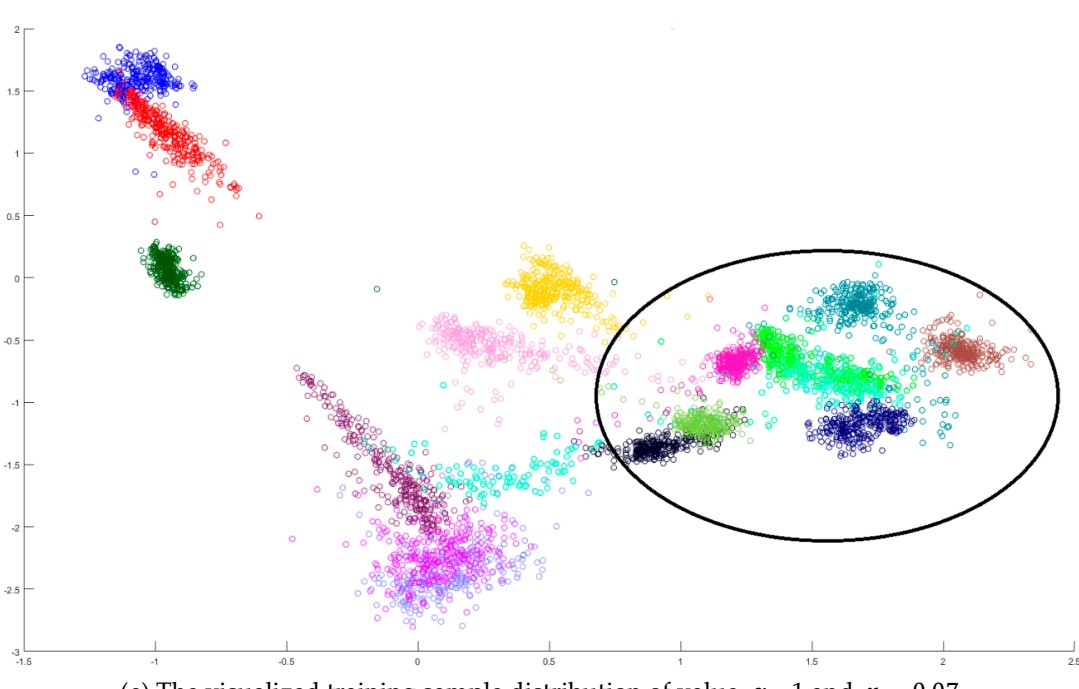

(**e**) The visualized training sample distribution of value $\alpha = 1$ and $r = 0.07$.

**Figure 4.** The analysis for the training data in dataset Salinas: (**a**) The overlapping degrees of various values $\alpha$, (**b**) The visualized training sample distribution of PCA and $r = 0.41$, (**c**) The visualized training sample distribution of value $\alpha = 0$ and $r = 0.34$, (**d**) The visualized training sample distribution of value $\alpha = 0.95$ and $r = 0.18$, and (**e**) The visualized training sample distribution of value $\alpha = 1$ and $r = 0.07$.

After determining value $\alpha = 1.0$, the whole dataset of 54,129 samples was tested for evaluation. To show the robustness of the proposed method, the experiments were run 30 times for all methods, and the averaged rates were obtained for the comparison as shown in Table 4. These results are represented in the averaged accuracy rates plus/minus the standard derivation form. The classification results of three RD algorithms, PCA, PCA plus FLE, and PCA plus SVMFLE, are tabulated in Table 4 ((a) The NN classifier) using the

NN classifier. In addition to the NN classifier, a GAN classifier was also trained by using the reduced data for the comparison with Zhu's work [10]. For a fair comparison, the GAN classifier was trained by the program whose source codes were accessed from [10]. The same architecture of GAN with the same initial weights was trained. A total of 200 training samples of dimensions 5 were randomly chosen for training the GAN classifier, and the remaining 53,929 samples were tested. Here, the training epochs were set as 100. The classification results of various RD algorithms are listed in Table 4 ((b) The GAN classifier) using the GAN classifier. From the classification results as listed in Table 4, the proposed SVMFLE-based DR method outperforms the other methods. Moreover, the standard derivations of SVMFLE are also smaller than the other methods. We can claim that the SVMFLE method provides the DR transformation matrix with more discriminant power than the other methods. To visualize the classification results, the classified label maps for dataset Salinas are presented as shown in Figure 5. Six algorithms, three DR methods times two classifiers, (PCA, NN), (PCA + FLE, NN), (PCA + SVMFLE, NN), (PCA, GAN) (Zhu's method), (PCA + FLE, GAN), and (PCA + SVMFLE, GAN), were performed for the comparison. Two elements in the brackets represent the DR method and the classifier, respectively. Observing the classified label maps, the proposed PCA + SVMFLE DR method plus the GAN classifier obtains fewer speckle-like errors than the other methods especially in classes Grapes_untrained, Vinyard_untrained, and Soil_vinyard_develop.

**Table 3.** Pixel numbers of dataset Salinas.

| No. | Colors | Class Labels | Sample # |
|:---:|:---:|:---:|:---:|
| 1. | | Brocoli_green_weeds_1 | 2009 |
| 2. | | Brocoli_green_weeds_2 | 3726 |
| 3. | | Fallow | 1976 |
| 4. | | Fallow_rough_plow | 1394 |
| 5. | | Fallow_smooth | 2678 |
| 6. | | Stubble | 3959 |
| 7. | | Celery | 3579 |
| 8. | | Grapes_untrained | 11,271 |
| 9. | | Soil_vinyard_develop | 6203 |
| 10. | | Corn_senesced_green_weeds | 3278 |
| 11. | | Lettuce_romaine_4wk | 1068 |
| 12. | | Lettuce_romaine_5wk | 1927 |
| 13. | | Lettuce_romaine_6wk | 916 |
| 14. | | Lettuce_romaine_7wk | 1070 |
| 15. | | Vinyard_untrained | 7268 |
| 16. | | Vinyard_vertical_trellis | 1807 |
| | Total | | 54,129 |

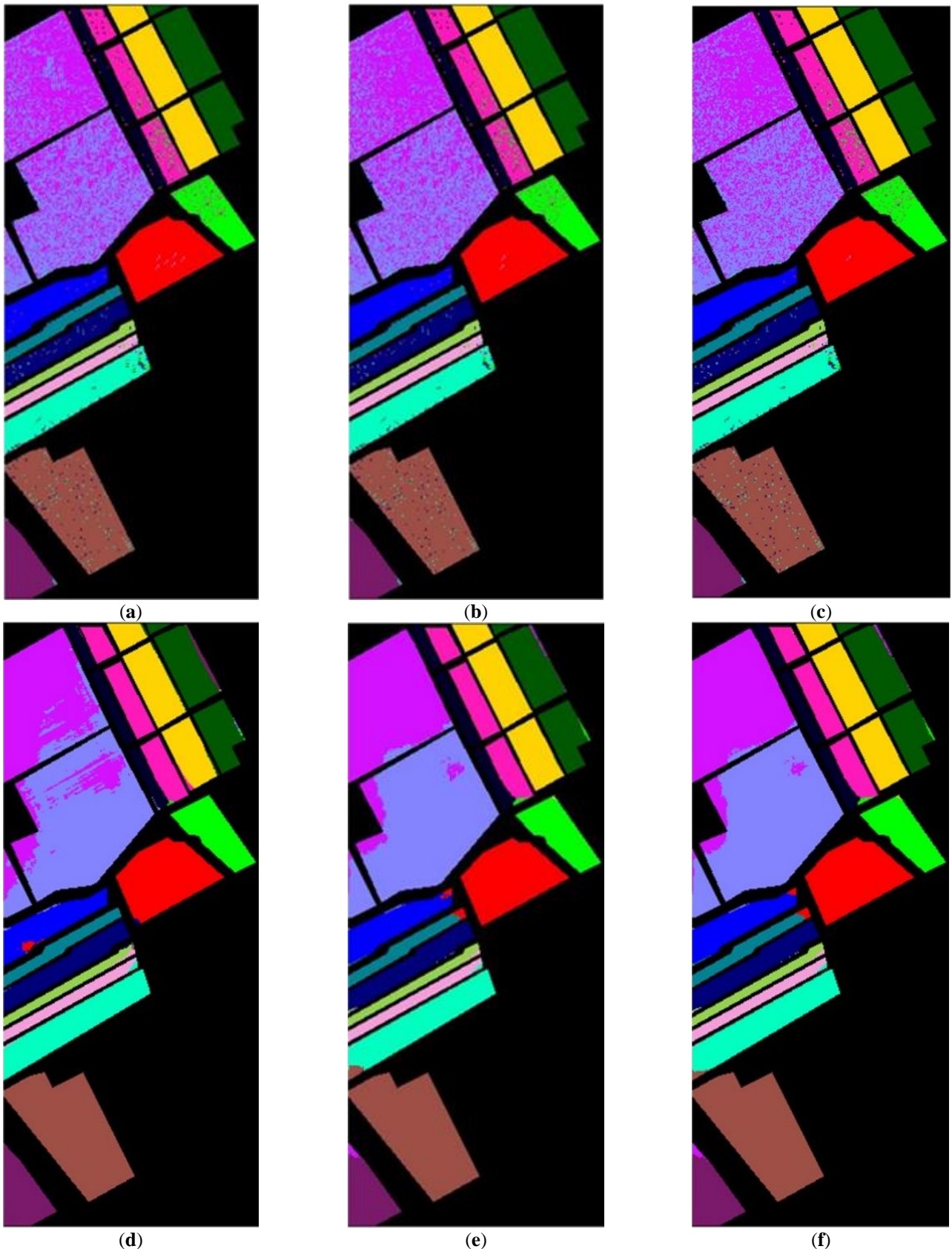

**Figure 5.** The classified label maps for dataset Salinas using three DM methods plus two classifiers. (**a**) (PCA, NN) (**b**) (PCA + FLE, NN), (**c**) (PCA + SVMFLE, NN), (**d**) (PCA, GAN) (Zhu's method), (**e**) (PCA + FLE, GAN), and (**f**) (PCA + SVMFLE, GAN).

**Table 4.** The classification results of various DR methods on dataset Salina.

| DR methods | PCA | PCA + FLE | PCA + SVMFLE ($\alpha = 1$) |
|---|---|---|---|
| OA (%) | $84.1 \pm 0.4$ | $85.1 \pm 0.5$ | $\mathbf{89.8 \pm 0.4}$ |
| AA (%) | $91.7 \pm 0.2$ | $93.0 \pm 0.3$ | $\mathbf{95.8 \pm 0.1}$ |
| K × 100 | $82.4 \pm 0.4$ | $84.0 \pm 0.5$ | $\mathbf{88.6 \pm 0.5}$ |

(a)   The NN classifier

| DR methods | PCA (Zhu's method) | PCA + FLE | PCA + SVMFLE ($\alpha = 1$) |
|---|---|---|---|
| OA (%) | $92.8 \pm 2.2$ | $94.1 \pm 1.8$ | $\mathbf{96.3 \pm 1.1}$ |
| AA (%) | $93.6 \pm 2.1$ | $94.9 \pm 2.1$ | $\mathbf{95.9 \pm 1.9}$ |
| K × 100 | $92.1 \pm 2.4$ | $93.4 \pm 2.0$ | $\mathbf{95.8 \pm 1.2}$ |

(b)   The GAN classifier

### 4.3. Classification Results of Dataset Pavia University

In this sub-section, the Pavia University dataset was used to evaluate the performance of the proposed SVMFLE DR algorithm. Several state-of-the-art algorithms were also compared to show effectiveness. The Pavia University HSI was captured from the Reflective Optics System Imaging Spectrometer (ROSIS) instrument and covered Pavia City, Italy. Nine land-cover classes covered 610 by 340 pixels with 103 spectral bands from 0.43 to 0.86 um. The false-color image and class label map are displayed in Figure 6, and their corresponding pixel numbers for all classes are listed in Table 5. In the experiment, all samples were used to generate the PCA feature space, for example, the same DR process in Zhu's method. After the projection, 180 reduced samples of each class, for example, 2700 samples, were randomly selected to generate the SVMFLE-based transformation matrix. Similar to the process in dataset Salinas, the dispersion indices were calculated for various values $\alpha$ as shown in Figure 7a, and the distributions of reduced samples are also displayed on a 2D plane in Figure 7b–e). The best value $\alpha = 0.76$ is selected according to the smallest dispersion index $r = 0.14$, which will be used in Equation (10). After determining value $\alpha = 0.76$, 42,776 pixels were tested to evaluate the performance. The experiments were executed 30 times to calculate the average accuracy rates. The accuracy rates for various DR methods, PCA, FLE, and SVMFLE, are listed in Table 6 ((a) The NN classifier) by using the NN classifier. Similarly, the reduced features are also classified by a GAN classifier. 200 training samples of dimensions 5 were randomly chosen for training the GAN classifier. The rest 42,576 samples in the dataset were tested for evaluation. Here, 100 epochs were performed during the training process. The classification results using a GAN classifier are shown in Table 6 ((b) The GAN classifier).

The proposed SVMFLE DR method outperforms the other methods from the classification results as shown in Table 6. Similar to the experiments on dataset Salina, all pixels in dataset Pavia University were projected into a five-dimensional feature space by the linear transformation $y_i = W^T x_i$ trained by the DR methods. Next, 200 five-dimensional samples were also randomly selected to train the NN and GAN classifiers. The discriminator in GAN was referred to as the classifier. The classified label maps for dataset Pavia University were generated by six algorithms, three DR methods, and two classifiers, as shown in Figure 8. They include three DR methods and two classifiers, for example, (PCA, NN), (PCA + FLE, NN), (PCA + SVMFLE, NN), (PCA, GAN), (PCA + FLE, GAN), and (PCA + SVMFLE, GAN). The proposed SVMFLE DR cooperated with a GAN classifier obtains fewer speckle-like errors than the other methods, especially in classes Meadows, Bare soil, and Asphalt. A better feature space is obtained for better classification using the GAN classifier.

**Table 5.** Numbers of pixels on dataset Pavia University.

| No. | Colors | Class Labels | Sample # |
|---|---|---|---|
| 1. | | Asphalt | 6631 |
| 2. | | Meadows | 18,649 |
| 3. | | Gravel | 2099 |
| 4. | | Trees | 3064 |
| 5. | | Painted metal sheets | 1345 |
| 6. | | Bare Soil | 5029 |
| 7. | | Bitumen | 1330 |
| 8. | | Self-Blocking Bricks | 3682 |
| 9. | | Shadows | 947 |
| | Total | | 42,776 |

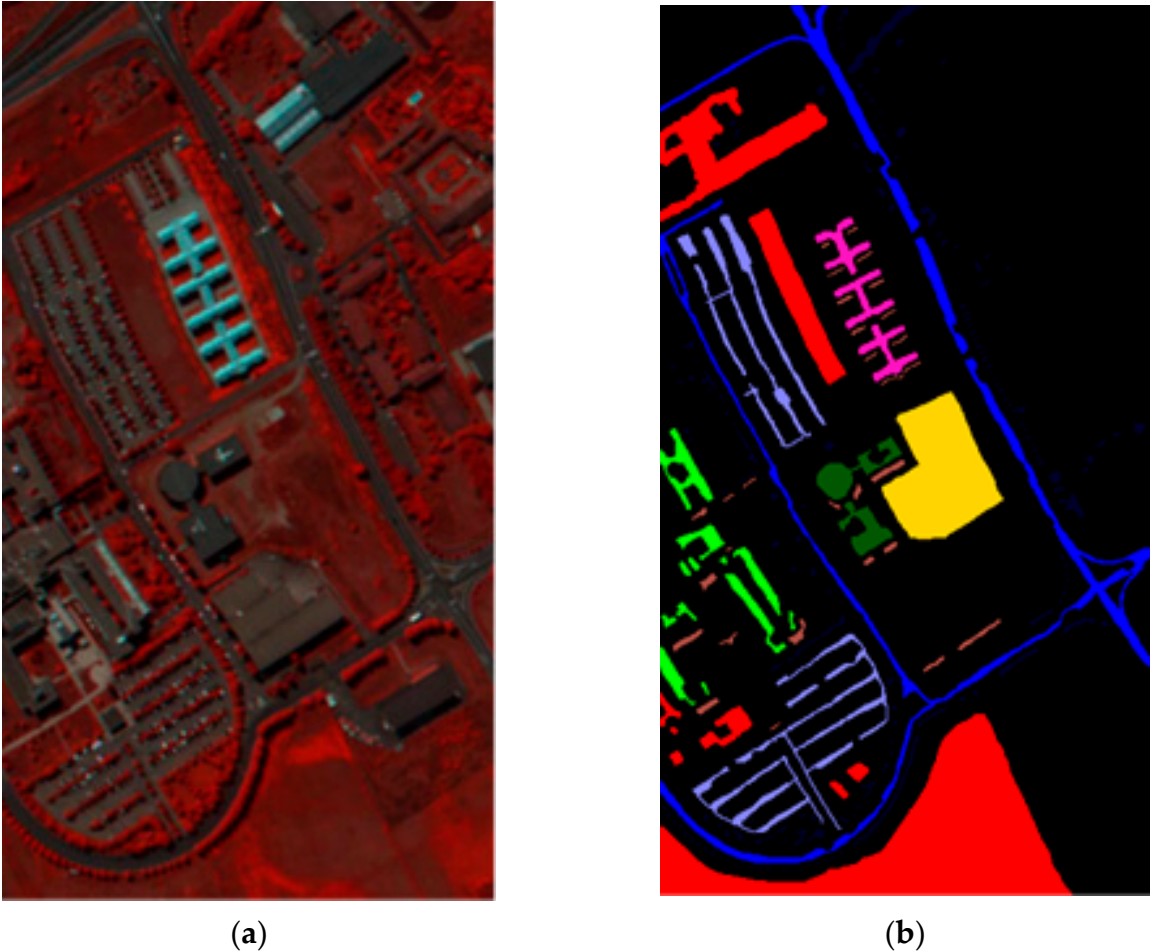

(**a**)    (**b**)

**Figure 6.** Dataset Pavia University: (**a**) False color image; (**b**) Class label map.

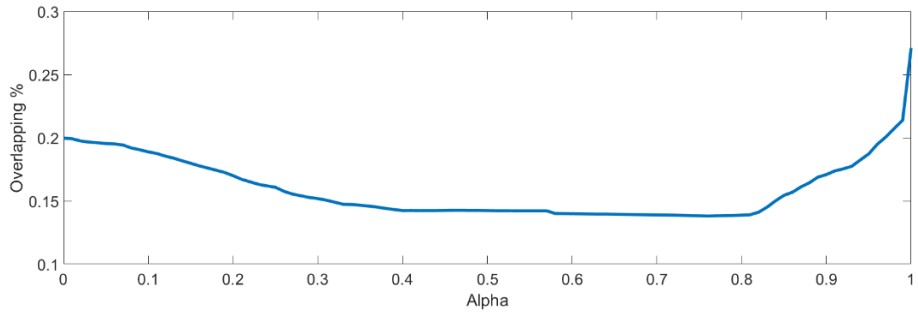

(**a**) The overlapping degrees of various values α.

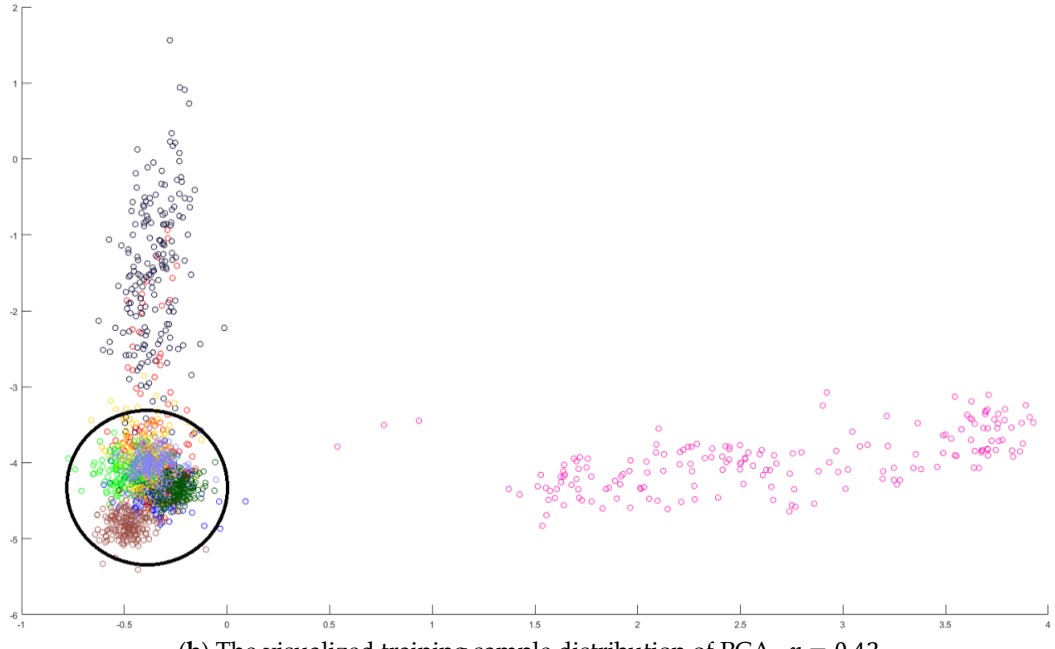

(**b**) The visualized training sample distribution of PCA, $r = 0.42$.

(**c**) The visualized training sample distribution of value $\alpha = 0$, $r = 0.21$.

**Figure 7.** *Cont.*

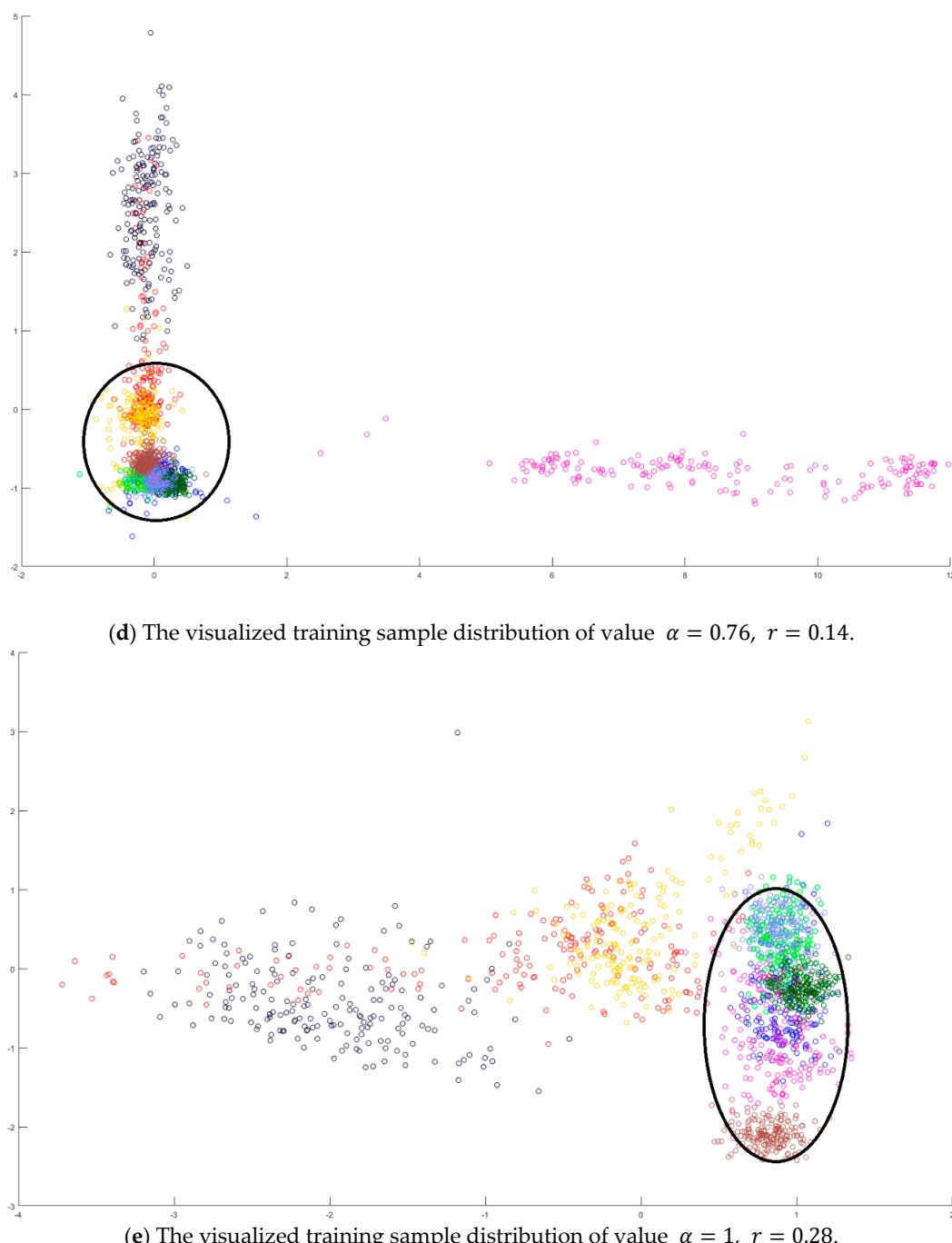

(**d**) The visualized training sample distribution of value $\alpha = 0.76$, $r = 0.14$.

(**e**) The visualized training sample distribution of value $\alpha = 1$, $r = 0.28$.

**Figure 7.** The analysis for the training data in dataset Pavia University: (**a**) The overlapping degrees of various values $\alpha$, (**b**) The visualized training sample distribution of PCA and $r = 0.42$, (**c**) The visualized training sample distribution of value $\alpha = 0$ and $r = 0.21$, (**d**) The visualized training sample distribution of value $\alpha = 0.76$ and $r = 0.14$, and (**e**) The visualized training sample distribution of value $\alpha=1$ and $r = 0.28$.



**Table 6.** The classification results on dataset Pavia University.

| DR methods | PCA | PCA + FLE | PCA + SVMFLE ($\alpha = 0.76$) |
|---|---|---|---|
| OA | $40.1 \pm 2.5$ | $82.8 \pm 2.4$ | **$86.0 \pm 1.1$** |
| AA | $58.2 \pm 2.1$ | $86.3 \pm 0.9$ | **$88.2 \pm 0.2$** |
| K $\times$ 100 | $29.7 \pm 2.6$ | $79.2 \pm 2.9$ | **$81.8 \pm 1.3$** |
| | (a) | The NN classifier | |
| DR methods | PCA (Zhu's method) | PCA + FLE | PCA + SVMFLE ($\alpha = 0.76$) |
| OA (%) | $87.5 \pm 1.5$ | $88.5 \pm 1.7$ | **$89.2 \pm 1.8$** |
| AA (%) | $72.7 \pm 5.1$ | $74.9 \pm 5.3$ | **$75.8 \pm 5.2$** |
| K $\times$ 100 | $83.2 \pm 2.1$ | $84.6 \pm 2.3$ | **$85.5 \pm 2.5$** |
| | (b) | The GAN classifier | |

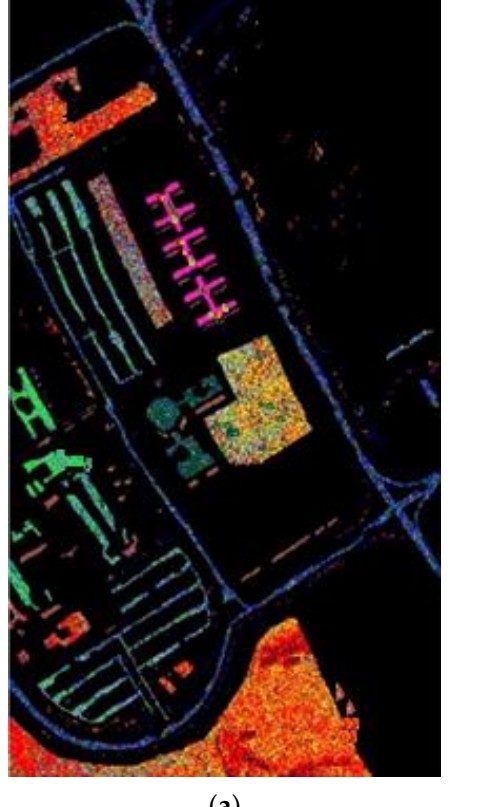
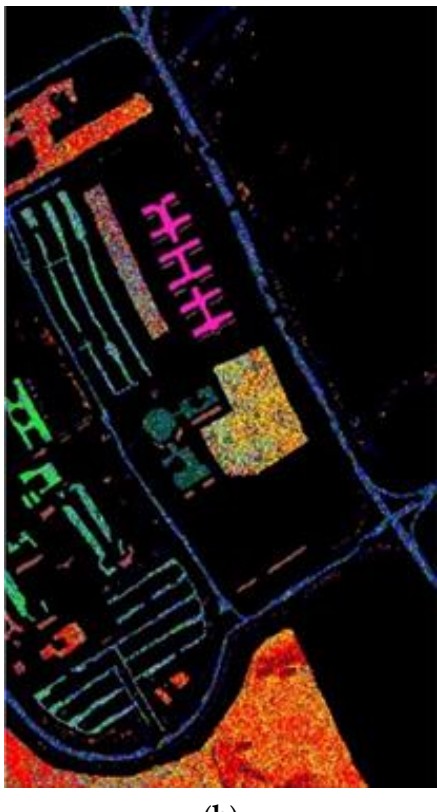

(**a**)　　　　　　　　　　　　　　　　(**b**)

**Figure 8.** *Cont.*

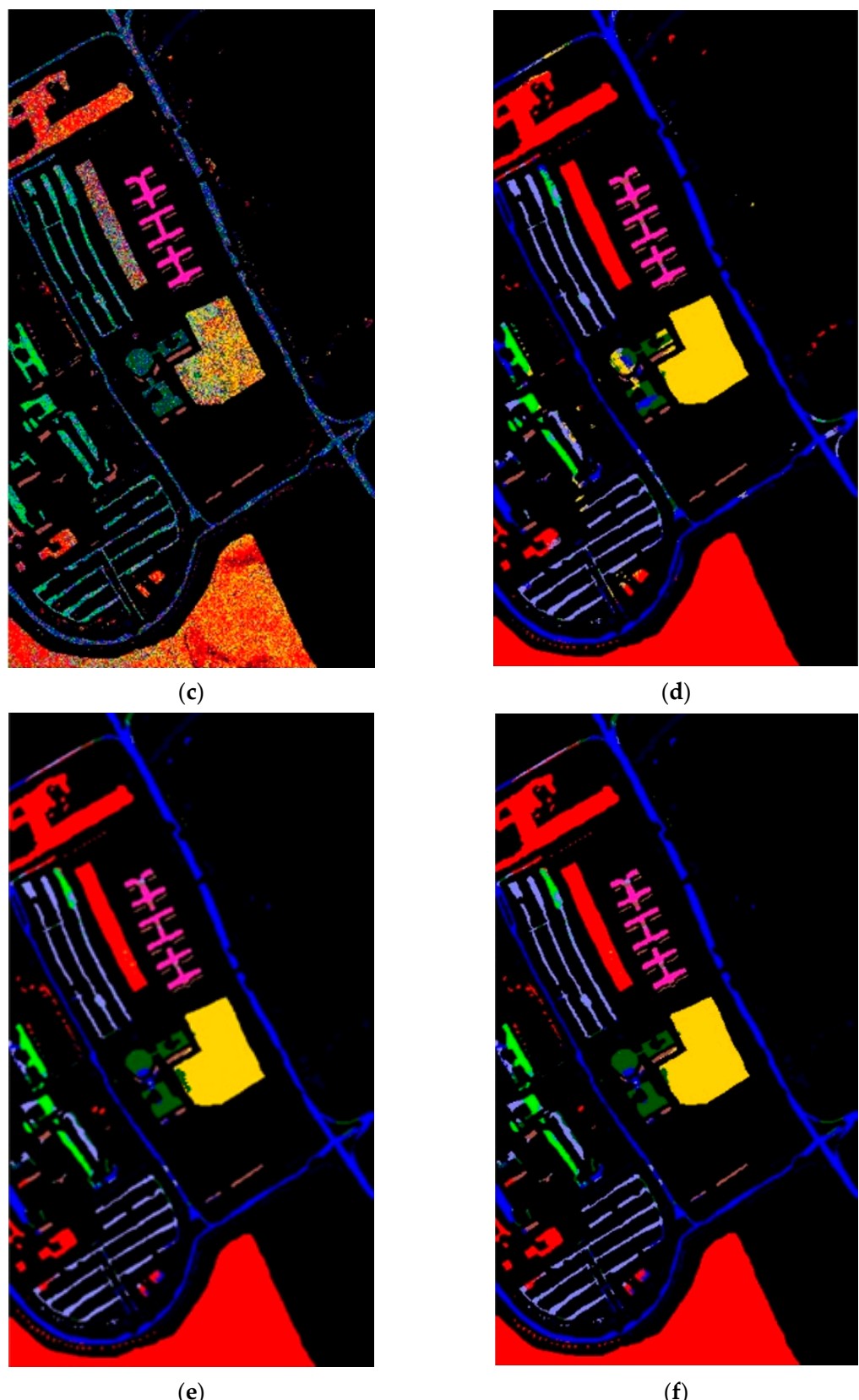

**Figure 8.** The classification label maps of dataset Pavia University. (**a**) (PCA, NN) (**b**) (PCA + FLE, NN), (**c**) (PCA + SVMFLE, NN), (**d**) (PCA, GAN), (**e**) (PCA + FLE, GAN), and (**f**) (PCA + SVMFLE, GAN).

*4.4. Classification Results of Dataset Indian Pines Site(IPS)*

In this sub-section, the Indian Pines Site (IPS) dataset was used to evaluate the effectiveness of the proposed method and compared with the state-of-the-art methods. The IPS image was captured from AVIRIS, which was constructed by the Jet Propulsion Laboratory and NASA/Ames in 1992. The area from six miles in the western area of Northwest Tippecanoe County (NTC) was scanned to obtain this HSI dataset. The false-color image and its corresponding class label map are shown in Figure 9. A total of 16 land-cover classes of 145 × 145 pixels in 224 spectral bands are manually labeled. The sample numbers of each class are tabulated in Table 7. After removing the bands covering the water absorption, in total, 10,249 labeled pixels of 200 spectral bands are used in the experiments. In the experiments, three DR methods (e.g., PCA, FLE, and SVMFLE) and two classifiers (e.g., NN and GAN) are implemented for performance comparison. Similar to the experimental configurations on datasets Salina and Pavia University, PCA is first performed to obtain the PCA feature space. Since the samples in each class are biased, the training samples for SVMFLE DR training are selected by the following rule. 300 training samples in a specified class were randomly chosen for obtaining the SVMFLE transformation matrix if its sample number is larger than 300; otherwise, 75% of samples are randomly selected for training. The specified training sample numbers are tabulated at the rightmost column in Table 7.

**Table 7.** Numbers of pixels on dataset IPS.

| No. | Colors | Class Labels | Sample # in Class | Sample # for Training |
|-----|--------|--------------|-------------------|----------------------|
| 1. | | Alfalfa | 46 | 35 |
| 2. | | Corn-notill | 1428 | 300 |
| 3. | | Corn-min | 830 | 300 |
| 4. | | Corn | 237 | 178 |
| 5. | | Grass-pasture | 483 | 300 |
| 6. | | Grass-trees | 730 | 300 |
| 7. | | Grass-pasture-mowed | 28 | 21 |
| 8. | | Hay-Windrowed | 478 | 300 |
| 9. | | Oats | 20 | 15 |
| 10. | | Soybean-notill | 972 | 300 |
| 11. | | Soybean-mintill | 2455 | 300 |
| 12. | | Soybean-clean | 593 | 300 |
| 13. | | Wheat | 205 | 154 |
| 14. | | Woods | 1265 | 300 |
| 15. | | Buildings-Grass-Trees | 386 | 300 |
| 16. | | Stone-Steel-Towers | 93 | 70 |
| | Total | | 10,249 | 3473 |

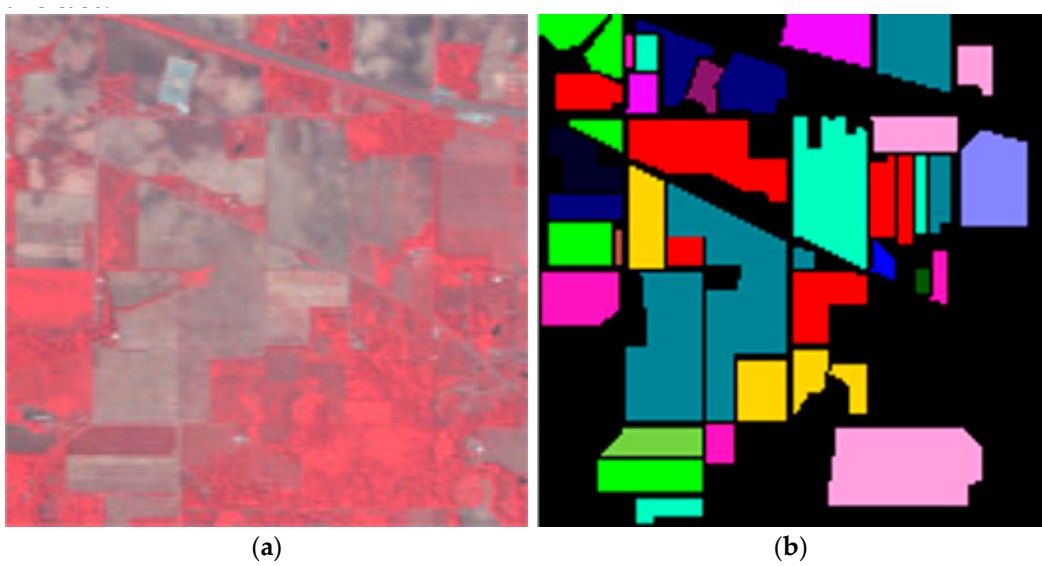

**Figure 9.** Dataset IPS: (**a**) False color image; (**b**) Class label map.

Next, value $\alpha$ in Equation (10) is also determined by the dispersion index from the training samples. It is varied from 0 to 1 in a 0.01 increasing step, and the corresponding dispersion index is calculated as shown in Figure 10a. The smallest value $r = 0.29$ is obtained when $\alpha = 0.74$. Similarly, all training samples were projected to 2D space, and the sample distributions versus various $\alpha$ and $r$ are displayed in Figure 10b–e. Observing the sample distributions drawn in the circles, the sample dispersion in Figure 10d is the largest with its corresponding values $r = 0.29$ and $\alpha = 0.74$. In a similar manner, the proposed SVMFLE DR method cooperated with the GAN classifier was also compared with Zhu's work [10]. A total of 200 samples of 5 dimensions were randomly chosen for the training of the GAN classifier, and the remaining 10,049 samples were tested by the trained GAN classifier. One hundred epochs were executed during the training process. In addition, the same network architecture with the same initial weights was set for various compared methods. The experiments were run 30 times to obtain the averaged accuracy rates and the standard derivations for the fair comparisons. The classification results are tabulated in Table 8 by using PCA, FLE, and SVMFLE DR methods cooperated with two classifiers. From the results in Table 8, the proposed SVMFLE DR method outperforms all the other two DR methods both using the NN and GAN classifiers.

**Table 8.** The classification results on dataset IPS.

| 1. DR method | 2. PCA | 3. PCA + FLE | 4. PCA + SVMFLE ($\alpha = 0.74$) |
|---|---|---|---|
| 5. OA | 6. $59.4 \pm 1.1$ | 7. $74.0 \pm 0.8$ | 8. $\mathbf{76.2 \pm 0.3}$ |
| 9. AA | 10. $76.1 \pm 0.6$ | 11. $84.4 \pm 0.5$ | 12. $\mathbf{85.6 \pm 0.5}$ |
| 13. K × 100 | 14. $54.8 \pm 1.2$ | 15. $71.3 \pm 0.9$ | 16. $\mathbf{73.3 \pm 0.3}$ |
| | (a) The NN classifier | | |
| 17. DR method | 18. PCA (Zhu's method) | 19. PCA + FLE | 20. PCA + SVMFLE ($\alpha = 0.74$) |
| 21. OA | 22. $86.6 \pm 1.2$ | 23. $86.5 \pm 1.3$ | 24. $\mathbf{87.0 \pm 1.8}$ |
| 25. AA | 26. $72.8 \pm 6.5$ | 27. $72.3 \pm 5.8$ | 28. $\mathbf{73.5 \pm 6.1}$ |
| 29. K × 100 | 30. $84.1 \pm 1.4$ | 31. $84.0 \pm 1.5$ | 32. $\mathbf{85.6 \pm 2.0}$ |
| | (b) The GAN classifier | | |

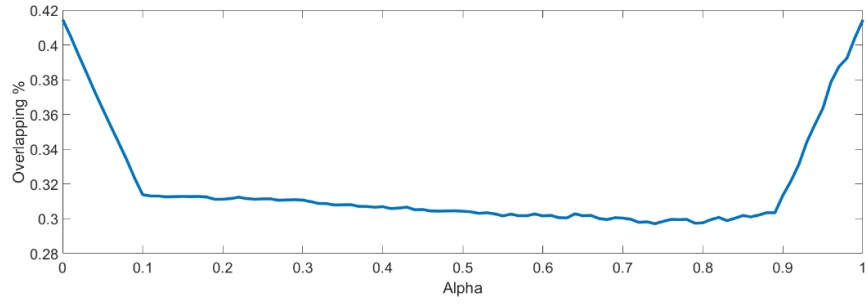

(**a**) The overlapping degrees of various values $\alpha$.

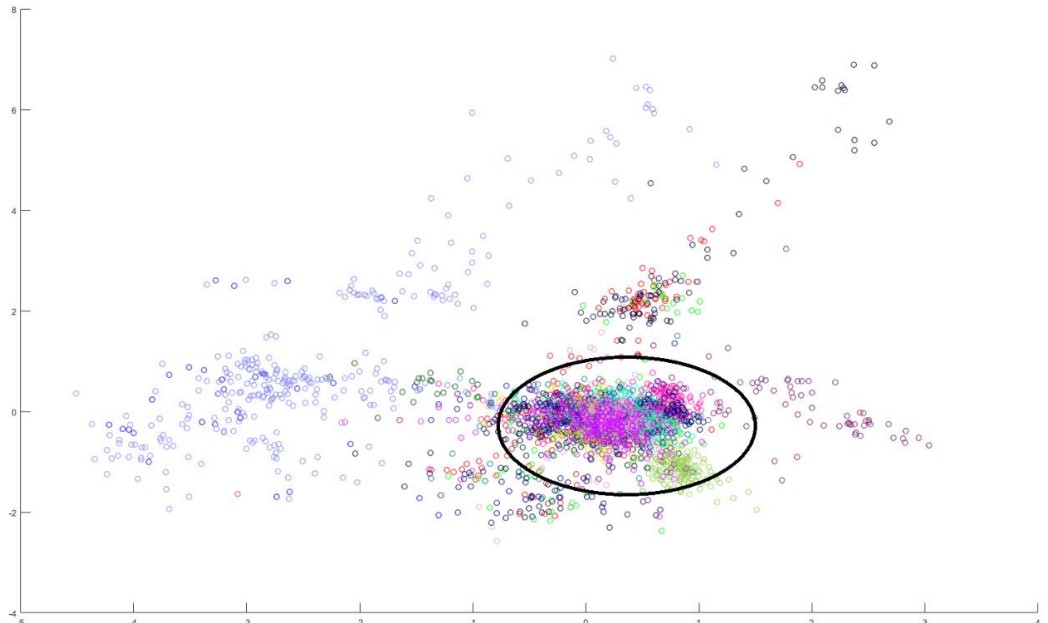

(**b**) The visualized training sample distribution of PCA, $r = 0.51$.

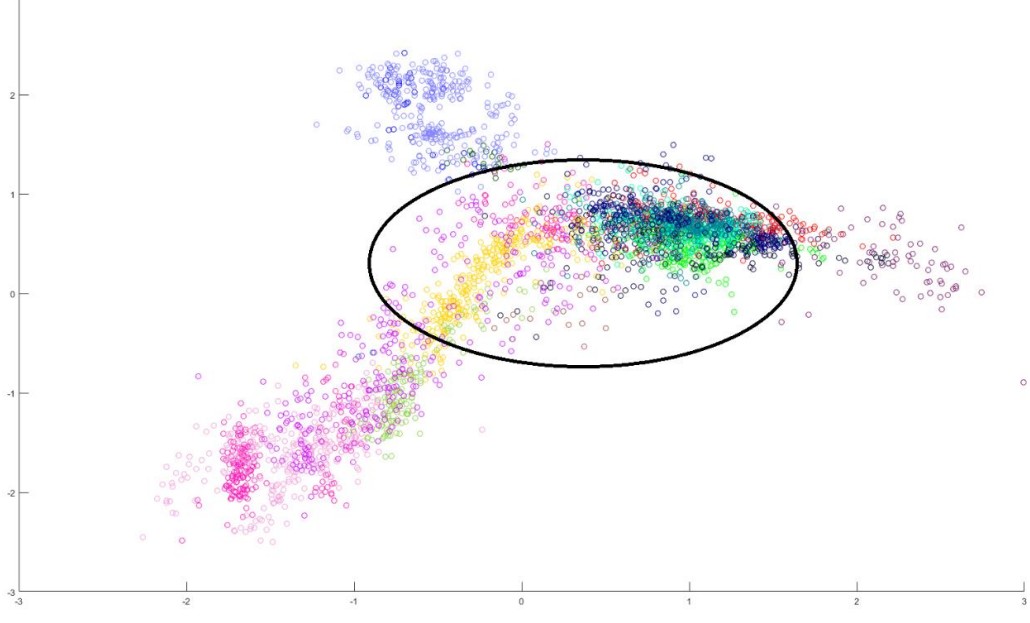

(**c**) The visualized training sample distribution of value $\alpha = 0$, $r = 0.42$.

**Figure 10.** *Cont.*

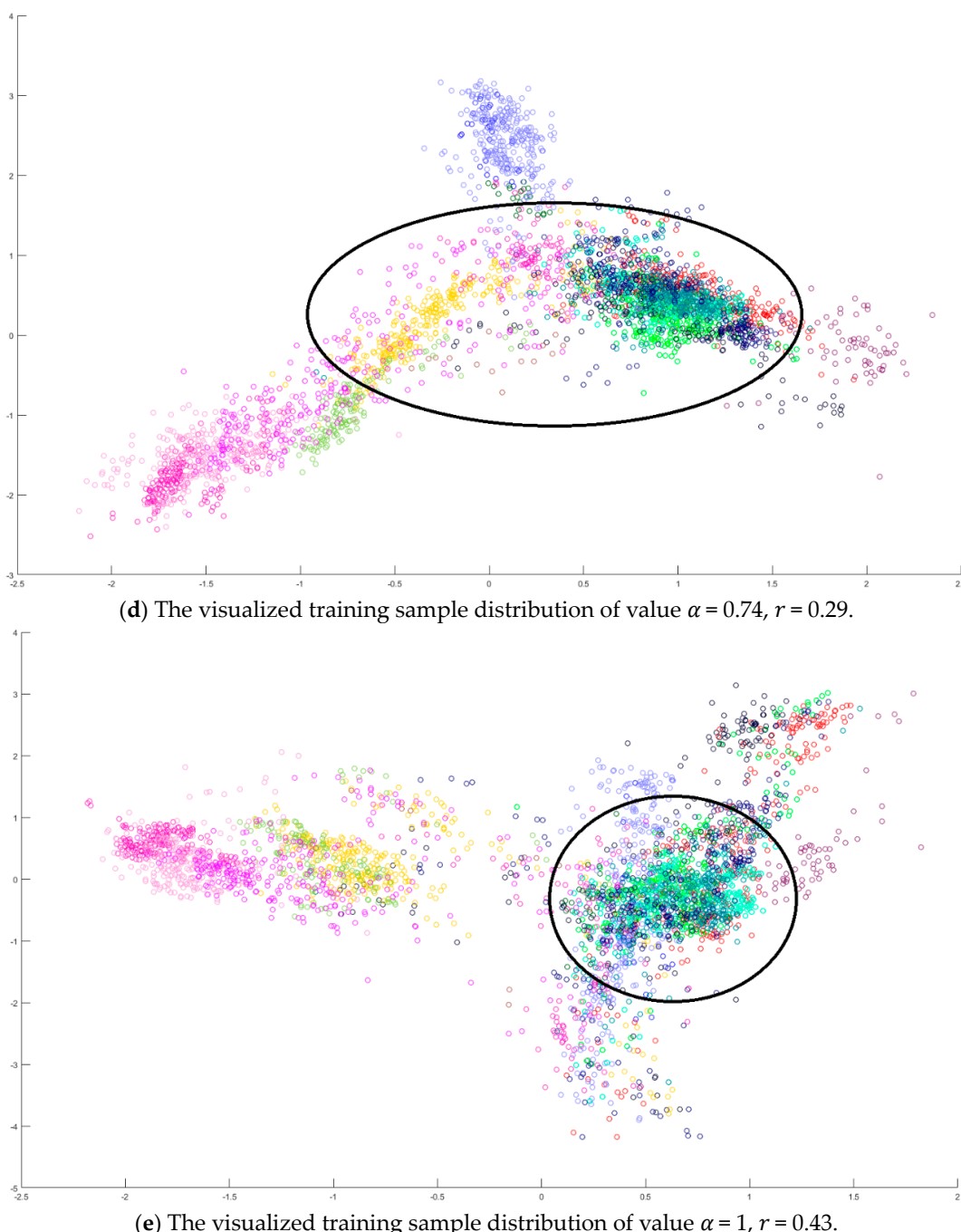

(**d**) The visualized training sample distribution of value $\alpha$ = 0.74, $r$ = 0.29.

(**e**) The visualized training sample distribution of value $\alpha$ = 1, $r$ = 0.43.

**Figure 10.** The analysis for the training data in dataset IPS: (**a**) The overlapping degrees of various values $\alpha$, (**b**) The visualized training sample distribution of PCA and $r$ = 0.51, (**c**) The visualized training sample distribution of value $\alpha$ = 0 and $r$ = 0.42, (**d**) The visualized training sample distribution of value $\alpha$ = 0.74 and $r$ = 0.29, and (**e**) The visualized training sample distribution of value $\alpha$ = 1 and $r$ = 0.43.

Figure 11 shows the classification maps of dataset IPS. Three DR methods and two classifiers were trained. In our opinion, the number of training samples for each class is biased. Therefore, the improvement of the proposed SVMFLE DR algorithm is limited in this case.

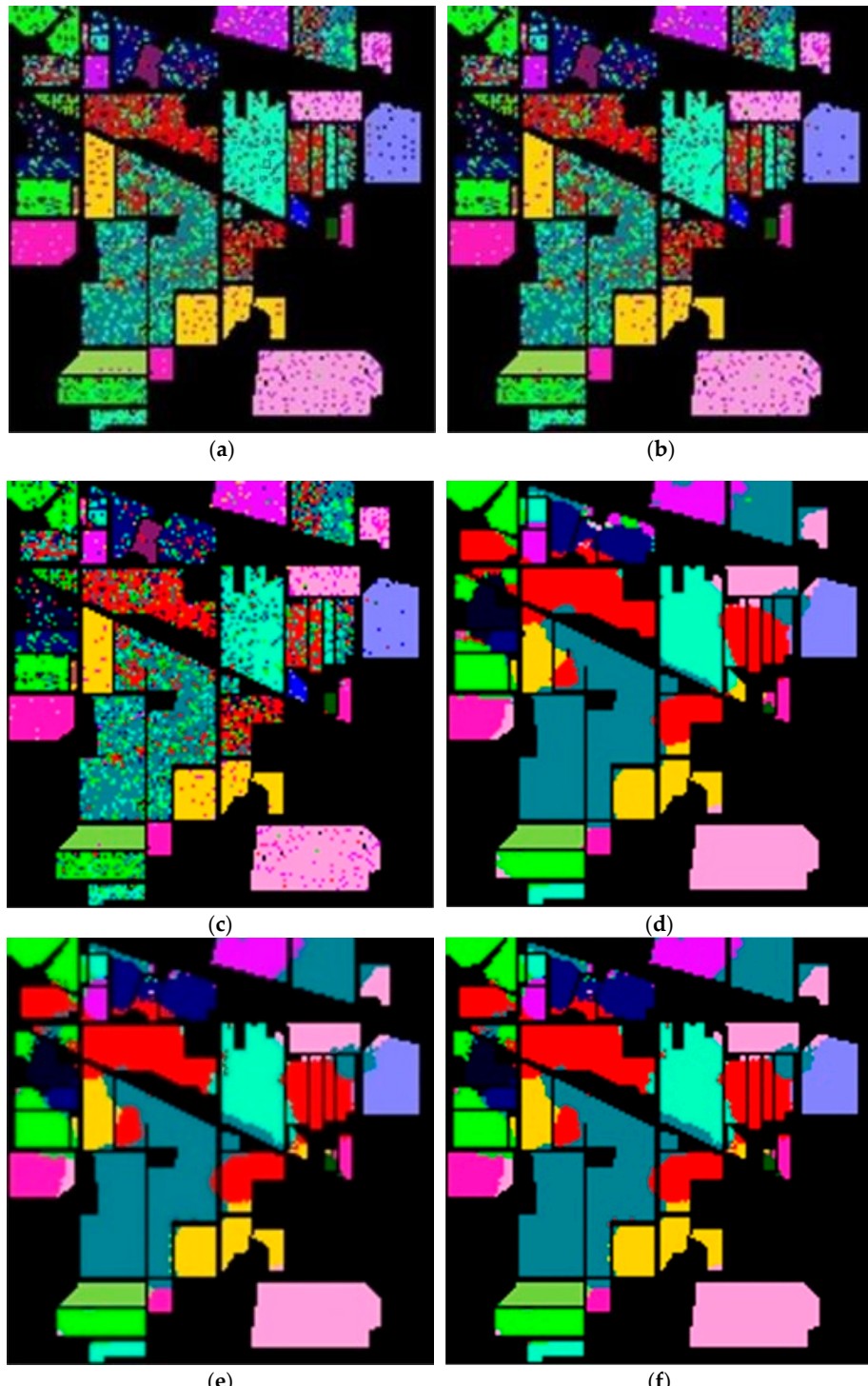

**Figure 11.** The classification label maps of dataset IPS. (**a**) (PCA, NN) (**b**) (PCA + FLE, NN), (**c**) (PCA + SVMFLE, NN), (**d**) (PCA, GAN), (**e**) (PCA + FLE, GAN), and (**f**) (PCA + SVMFLE, GAN).

The proposed SVMFLE with the GAN classifier method was employed in HSI classification. The SVMFLE algorithm applies the SVM selected samples to improve the representation of the between-scatter matrix, which could extract much more useful discrimination information. Accordingly, it obtained a better overall accuracy (OA) than that of Zhu's method to the improvements of 3.5%, 1.7%, and 0.4% for the Salinas, Pavia University, and Indian Pines Site datasets, respectively. In the meanwhile, the proposed SVMFLE with the GAN

classifier also obtained a better average accuracy (AA) than that of Zhu's method to the improvements of 2.3%, 3.1%, and 0.7% for these three datasets, respectively. Owing to the fact that dataset IPS greatly varies in the pixel numbers of each class, a biased classifier is easily trained due to the biased training samples. The proposed SVMFLE with the GAN classifier method outperforms Zhu's method only above 0.4% in OA. On the other hand, the pixel numbers of each class in dataset Salinas are large enough, the trained SVMFLE with the GAN classifier outperforms that of Zhu's method above 3.5% in OA. Therefore, when the training samples are large enough and unbiased, the proposed method can obtain a powerfully discriminant feature space and significantly improve the classification performance. Otherwise, the improvement is limited if the training samples are few and biased.

Finally, Figures 12 and 13 show the overall accuracy versus the reduced dimensionality in NN and GAN classifiers, respectively. From the results in Figures 12 and 13, the proposed SVMFLE DR method cooperated with the NN or GAN classifier outperforms the other methods. Only five principal components in the reduced feature space were chosen in considering the computational complexity and training time of the GAN classifier. In addition, the classification performance descended when dimensionality ascended (e.g., 40 components).

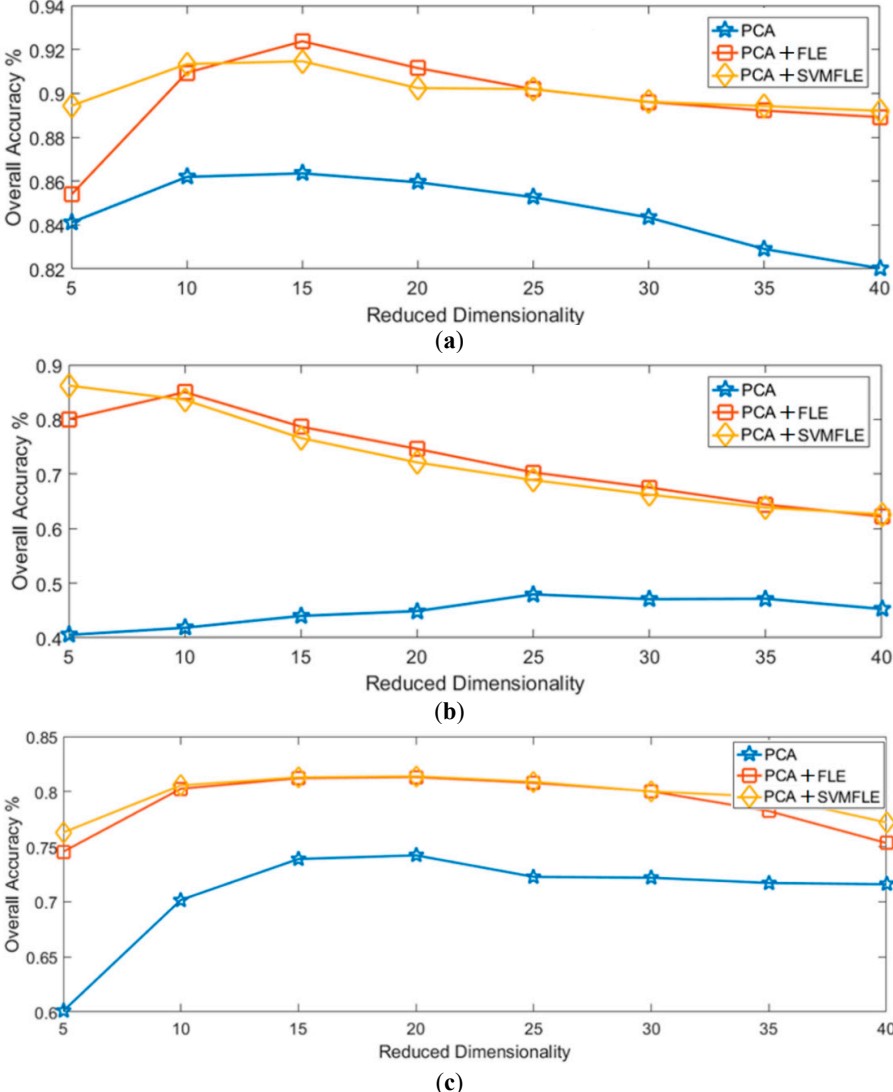

**Figure 12.** The reduced dimension versus the classification accuracy by NN classifier on three datasets: (**a**) Salinas; (**b**) Pavia University; (**c**) IPS.

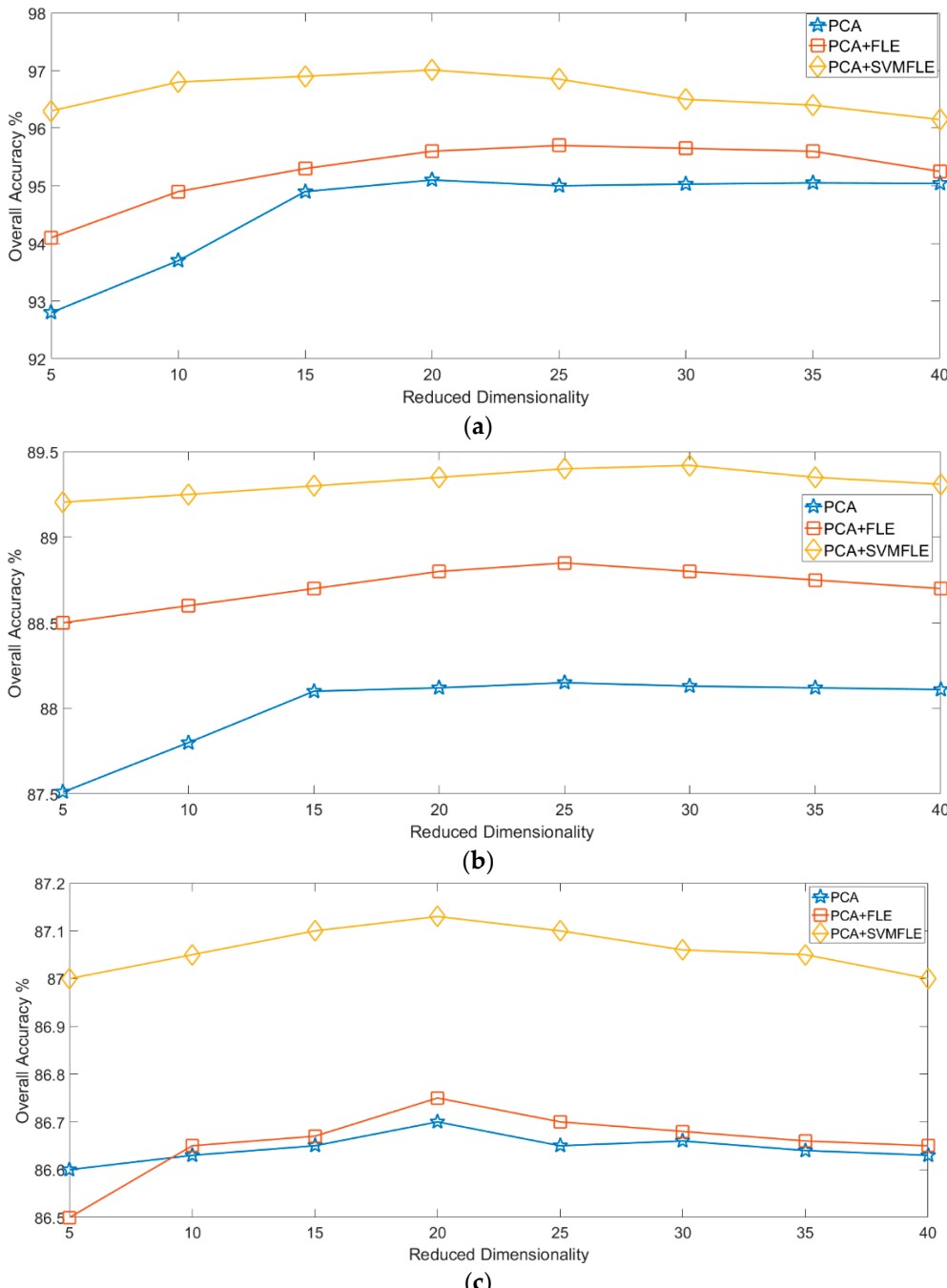

**Figure 13.** The reduced dimension versus the classification accuracy by GAN classifier on three datasets: (**a**) Salinas; (**b**) Pavia University; (**c**) IPS.

## 5. Conclusions

In this paper, we have proposed a DR scheme using the feature line embedding strategy with SVM selected samples. SVM selected training samples were used for calculating the between-class scatters. The dispersion degree among samples is calculated to automatically determine the better parameter $\alpha$ for $SSB_{SV}$. With the selected samples, a reduced space with more discriminant power is obtained. From the experimental results, the classifying ability of the classifier is improved in the reduced space. In addition, several state-of-the-art DR methods were implemented cooperated with the NN and GAN classifiers for the performance comparison. From the experimental results, the proposed

SVMFLE DR method outperforms the other methods in three performance indices. Accuracies of 96.3%, 89.2%, and 87.0% were obtained for the Salinas, Pavia University, and Indian Pines Site datasets using the GAN classifier, respectively. On the other hand, this scheme with the NN classifier also achieves 89.8%, 86.0%, and 76.2% accuracy rates for these three datasets. The improvements of the proposed method are significant both in datasets Salinas and Pavia University because of the large and unbiased training samples. On the contrary, when the training samples of each class are biased and few in dataset IPS, the performance of the proposed method is not obviously improved. It will be the future work to mitigate the impact of biased training samples.

**Author Contributions:** Y.-N.C. conceived the project, conducted research, performed initial analyses, and wrote the first manuscript draft. T.T. performed the initial analyses, advanced analyses, and edited the first manuscript. C.-C.H. edited the first manuscript and finalized the manuscript for communication. T.-J.L. performed analyses. K.-C.F. edited the first manuscript. All authors have read and agreed to the published version of the manuscript.

**Funding:** The work was funded by supported by the Ministry of Science and Technology under grant no. MOST 109-2221-E-239-026 and MOST 109-2634-F-008-008.

**Data Availability Statement:** The classification datasets are three public and avaliable datasets accessed from the following website http://www.ehu.eus/ccwintco/index.php/Hyperspectral_Remote_Sensing_Scenes.

**Conflicts of Interest:** The authors declare no conflict of interest.

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
