# Peer review of "Feature Line Embedding Based on Support Vector Machine for Hyperspectral Image Classification"

_remotesensing, doi:10.3390/rs13010130_

Round 1

Reviewer 1 Report

Dear Authors,

After careful review of your paper, I can note that this is a sound research with value for the readers. There are some minor to medium improvements necessary that would improve your paper.

  1. Introduction

Small typo Line 138 Section4, missing space.

  1. Related works

There should be no text between main section title (2.Related works) and subtitle (2.1. Feature Line Embedding (FLE)).

More precisely the text from Line 141 to Line 148 should be moved to the subtitle 2.1., or to the Introduction section (with minor changes in order to fit the text contextually).

  1. Feature Line Embedding based on Support Vector Machine (SVMFLE)

Line 237 Figure 2 has two parts/graphs. The first part is on the previous page. It is recommended to note in the Figure 2 caption that there are (a) and (b) parts/graphs (similarly as in Figure 5).

Overall, Tables and Figures should be noted in text closely to them and not a paragraph later. For example: You mention Table 2 in the text at Line 254, while the Table itself is at line 272. The sentence from 254 could be moved to line 271. Please revise this for every Table and Figure where applicable.

  1. Experimental Results

There should be no text between main section title (4. Experimental Results) and subtitle (4.1. Measurement Metrics).

The text from Line 275 to Line 277 can be removed, moved to previous  section or next subsection (with minor changes in order to fit the text contextually).

Figure 4 has multiple parts/graphs. This should be noted in the Figure caption (similarly as in Figure 5).

Figure 7 has multiple parts/graphs. This should be noted in the Figure caption (similarly as in Figure 5).

Figure 10 has multiple parts/graphs. This should be noted in the Figure caption (similarly as in Figure 5).

Captions for Table 7. and 8. are not in the same style as previous Table captions.

  1. Conclusions

Limitations, implications and contribution of the paper should be addressed. In addition, briefly note what can be conducted in future studies in this domain.

Kind regards,

Reviewer

Reviewer 2 Report

This manuscript does an interesting demonstration for feature line embedding based on support vector machine for hyperspectral image classification. This manuscript provides some exciting insight into hyperspectral image classification. In this manuscript, a novel feature line embedding algorithm proposed which is based on support vector machine, referred to as SVMFLE, for dimension reduction and for improving the performance of generative adversarial network in hyperspectral image classification.

 Introduction is very good, however authors should explain what new and original this paper has to offer beyond the already existing in the literature. What makes this work different from others? It should be better explained. In the last paragraph of the introduction, authors used a format which is good for a report or proposal not for a journal paper. “The rest of this paper is organized as follows: Previous and related works are discussed in Section 2. The proposed algorithm of SVM-based sample selection incorporated into the FLE is introduced in Section 3. The results demonstrating the effectiveness of the proposed method are presented in Section 4; the proposed method was compared with other state-of-the-art schemes for HSI classification to demonstrate its effectiveness. Finally, in Section 5, the conclusions are drawn.” Therefore, instead of using in Section 1 or 2, 5, a better paragraph needs to be provided.

During the reading of the manuscript, the following questions and comments came to my mind and I would like to ask the authors to comment on them:

  • In abstract will be good if more results be presented than some general finding.
  • In introduction is very good if authors explain what is new and original on this paper
  • In page 9 line 273, Fig 2 for each part A and B need to have explanation in caption then will be easier for reader to understand.
  • In page 13 line 330, Fig 4 for each part A, B, C, D and E need to have explanation in caption then will be easier for reader to understand.
  • In page 19 line 370, Fig 7 for each part A, B, C, D and E need to have explanation in caption then will be easier for reader to understand.
  • In page 24 line 414, Fig 10 for each part A, B, C, D and E need to have explanation in caption then will be easier for reader to understand.
  • In conclusion will be good if more results (actual finding supported by numbers) be presented than some general finding.

In summary, the reviewed article in current form could be accepted after a minor revision. Although the article provides a good body of work, suggested inclusion of additional information and modifications/corrections will definitely improve the standard and quality of the article.
